# High-performance cryo-temperature ionic thermoelectric liquid cell developed through a eutectic solvent strategy

Shuaihua Wang[1,2], Yuchen Li[1,2], Mao Yu[1], Qikai Li[1], Huan Li[1], Yupeng Wang[1], Jiajia Zhang[1], Kang Zhu[1] & Weishu Liu ◉[1] ✉

Ionic thermoelectric (i-TE) liquid cells offer an environmentally friendly, cost effective, and easy-operation route to low-grade heat recovery. However, the lowest temperature is limited by the freezing temperature of the aqueous electrolyte. Applying a eutectic solvent strategy, we fabricate a high-performance cryo-temperature i-TE liquid cell. Formamide is used as a chaotic organic solvent that destroys the hydrogen bond network between water molecules, forming a deep eutectic solvent that enables the cell to operate near cryo temperatures (down to −35 °C). After synergistic optimization of the electrode and cell structure, the as-fabricated liquid i-TE cell with cold (−35 °C) and hot (70 °C) ends achieve a high power density (17.5 W m$^{-2}$) and a large two-hour energy density (27 kJ m$^{-2}$). In a prototype 25-cell module, the open-circuit voltage and short-circuit current are 6.9 V and 68 mA, respectively, and the maximum power is 131 mW. The anti-freezing ability and high output performance of the as-fabricated i-TE liquid cell system are requisites for applications in frigid regions.

Ionic thermoelectric (i-TE) cells have attracted increasing attention because they provide high thermopower at low cost and can be easily operated. Therefore, they can potentially utilize low-grade heat (<100 °C) to power internet-of-things (IoT) devices[1,2]. The thermo-diffusion effects[3–8], thermogalvanic effects[9–15], and synergistic effects[16–18] in i-TE liquid cells and gel cells have been widely studied. In an i-TE liquid cell (sometimes called a thermogalvanic cell[10] or thermocell[19]), the thermopower derived by a temperature gradient ($S_i = -\frac{V_H - V_C}{T_H - T_C}$) is mainly contributed by the thermogalvanic effect since the coupling transport of thermodiffusive anion and cation cancels the statistic potential[16]. The convection endows redox couple a continuous conversion from heat to electricity in the liquid-state i-TE solutions, and the obvious advantage of large power output as compared with its counterpart of i-TE gel cell[20]. The power generation of i-TE cells is determined by the two-hour energy density ($E_{2h}$) and the instantaneous power $P_{max} = V_{oc}I_{sc}/4$, where $V_{oc}$ and $I_{sc}$ are the open-circuit

voltage and short-circuit current, respectively. The $P_{max}$ of i-TE liquid cells can be improved through three general strategies: (i) modifying the electrolyte to boost the $S_i$, which is related to the solvation-structural entropy difference ($\Delta S$) and the temperature-dependent concentration ratio difference ($\Delta C_r$) between the redox species[15,21–23], (ii) designing an electrode microstructure that accelerates electron exchange between the redox couple and electrode, which decides the $I_{sc}$[24–28], and (iii) extending the maximum working temperature range ($\Delta T_{max}$) to increase the $V_{oc}$ or $I_{sc}$ of an i-TE liquid cell[18]. Many researchers have modified the electrolyte to enhance the $\Delta S$ and consequently the $S_i$. Zhou et al. boosted the $S_i$ to 4.2 mV K$^{-1}$ by introducing urea and guanidinium (GdmCl) to an aqueous K$_3$Fe(CN)$_6$/K$_4$Fe(CN)$_6$ electrolyte (abbreviated as FeCN$^{4-/3-}$). They reported that guanidinium and urea synergistically reorganize the solvation structures of the redox couple[15]. Teppei et al. adopted an amphiphilic ion-pairing strategy that exploits the binding-affinity difference between

[1]Department of Materials Science and Engineering, Southern University of Science and Technology, Shenzhen, Guangdong 518055, China. [2]These authors contributed equally: Shuaihua Wang, Yuchen Li. ✉e-mail: liuws@sustech.edu.cn

dodecyl tetramethylammonium bromide micelles and the redox species, thus changing the $S_i$ of $FeCN^{4-/3-}$ from 1.4 to $-3.5\,mV\,K^{-1}$[21]. Meanwhile, enhancing the $\Delta C_r$ boosts the $S_i$ because the $\Delta C_r$ directly affects the concentration of the activated molecules required for the electrode reaction[22]. Employing a supramolecular approach that confines $I_3^-$ ions within $\alpha$-cyclodextrin cavities, Yamada et al. created an $I_3^-$ concentration gradient in an aqueous $I^-/I_3^-$ electrolyte that increased the $S_i$ from $-0.86$ to $-1.97\,mV\,K^{-1}$[23]. Zhou et al. devised a thermosensitive crystallization methodology in which the $FeCN^{4-}-Gdm^+$ complex is crystallized, precipitated, and dissolved to generate a sustained concentration gradient. Their methodology achieved an $S_i$ of $3.73\,mV\,K^{-1}$ in a 0.4 M $FeCN^{4-/3-}$ aqueous electrolyte[22].

Meanwhile, carbon-based nanomaterials with high specific surface areas have markedly advanced over the last decade. Baughman et al. designed a carbon nanotube-reduced graphene oxide composite electrode with adjustable porosity. They confirmed a positive correlation between the electroactive surface area of the electrode and the $P_{max}$ in the i-TE liquid cell[24]. Subsequent research has reported that various carbon-based electrodes, such as graphene aerogel[25], multiwalled carbon nanotubes[26], laser-etched PEDOT: PSS[27], and active carbon cloth[28], increase the $P_{max}$ by increasing the effective electrical conductivity ($\sigma_{eff}$) and $I_{sc}$. The $\Delta T_{max}$ is another critical parameter of i-TE cells. The up-limit temperature $T_{h,max}$ in an i-TE gel cell is determined by the thermal stability of the quasi-solid-state gel network, whereas the down-limit temperature $T_{c,min}$ is limited by the kinetic behavior of the thermodiffusing ions. Li et al. introduced glutaraldehyde to gelatin, forming strong covalent bonds that increased the $T_{h,max}$ from 30 °C to 44 °C while maintaining $T_{c,min}$ at 21 °C. They reported a high power density of $9.6\,mW\,m^{-2}\,K^{-2}$[18]. For an i-TE liquid cell, $T_{c,min}$ is limited by the freezing point of the solution[29]. The $T_{c,min}$ values of the reported i-TE liquid cells always exceed 0 °C. Specifically, the $T_{c,min}$ values of $I^-/I_3^-$ electrolyte, $FeCN^{4-/3-}$ electrolyte, $Fe^{2+}/Fe^{3+}$ electrolyte, and $Cu/Cu^{2+}$ electrolyte are 3 °C[23], 10 °C[25], 15 °C[13], and 20 °C[30], respectively. Depressing the $T_{c,min}$ into the cryo-temperature region remains a challenging task.

Benefiting from the natural of eutectic solvents that form a liquid mixture at a lower melting point than that of any individual component, electrolytes consisting of a eutectic mixture solvent such as Formamide (FA)/$H_2O$[31], Ethylene glycol (EG)/$H_2O$[32,33], and Dimethyl sulfoxide (DMSO)/$H_2O$[34], were promising candidates for cryogenic processes in energy-related systems. Given the ability of eutectic solvents to offer superior ionic transport conditions for electrolytes in subzero temperatures[35], they present a viable approach to lower the minimum operating temperature of ionic thermoelectric cells into the cryogenic range. For instance, Li et al. assemble a cryo-thermocell with a eutectic redox electrolyte of formamide and water and achieved a $P_{max}$ of $3.6\,W\,m^{-2}$ at $\Delta T = 106$ °C[25]. The present study explores the application of an $H_2O$/formamide (FA)-$FeCN^{4-/3-}$-GdmCl i-TE liquid cell at cryo temperatures. The cell is fabricated through a eutectic solvent strategy and synergistic optimization. We first introduce the chaotic organic FA solvent, which destroys the hydrogen bond network among the water molecules through molecular dipole interactions, forming a eutectic solvent with a low melting point. Next, we show that the hydrophilic properties of the porous carbon electrode critically determine the power output of the i-TE liquid cell. Finally, to enlarge the $\Delta T_{max}$, we investigate the thermal resistance effect of a thermally insulating separator (TIS) formed from cotton fibers. The instantaneous output power density ($P_{max}$) reached $17.5\,W\,m^{-2}$ and the two-hour energy density ($E_{2h}$) reached $27\,kJ\,m^{-2}$ between $-35$ °C at the cold end and 70 °C at the hot end. This work is expected to broaden the working temperature range and enhance the power output of i-TE cells, enabling their application in most human living areas, including frigid polar regions in the winter.

## Results and discussion

### Principle of high anti-freezing performance

Figure 1a illustrates how a eutectic solvent can decrease the $T_{c,min}$ of an i-TE liquid cell intended to operate at cryo temperatures. The freezing of the solvent limits the $T_{c,min}$ of the $H_2O$-$FeCN^{4-/3-}$-GdmCl liquid cell, suppressing the thermodiffusion of ions. Here we proposed a eutectic solvent strategy that decreases the freezing point ($T_f$) of water solvents. Theoretically, the $T_f$ of an electrolyte can be lowered by increasing the entropy of the solvent molecules, which decreases the Gibbs free energy of the liquid[36]. At the molecular level, water freezing involves the nucleation of ice from a random arrangement into a tetrahedrally coordinated structure through hydrogen bonding[37]. An additional solvent such as FA can form a solution that disturbs the hydrogen bond network between water molecules.

First, the $T_f$ of ionic solutions $H_2O$/$x$ FA-$y$ $FeCN^{4-/3-}$-$z$ GdmCl ($x = 0$–100 vol.% in 10-vol.% increments; $y = 0.4$ M; $z = 3$ M) were derived from differential scanning calorimetry (DSC) measurements (Supplementary Fig. 1). For simplicity, the ionic solutions $H_2O$/$x$ FA-0.4 M $FeCN^{4-/3-}$-3 M GdmCl are hereafter referred to as FA0–FA100, where the number denotes the volume percentage of FA. Figure 1b shows the phase diagram of the hybrid electrolyte determined from DSC measurements. The electrolyte exhibits obvious binary eutectic behavior. The $T_f$ of formamide-free FA0 ($-6$ °C) is below that of pure water (0 °C) because the $FeCN^{4-/3-}$ and GdmCl ions are solvated. In $H_2O$/$x$ FA-$y$ $FeCN^{4-/3-}$-$z$ GdmCl containing 0, 10, 20, 30, 40, 50, and 60 vol.% FA, the $T_f$ values are $-6$ °C, $-8$ °C, $-11$ °C, $-17$ °C, $-25$ °C, $-37$ °C, and $-52$ °C, respectively. This continuous decrease is followed by a $T_f$ increase at higher FA contents. The deepest eutectic point is $-52$ °C at an FA content of 60 vol.%.

The ionic conductivities of the ionic solutions were measured using electrochemical impedance spectroscopy (EIS) at different temperatures. The resistances yielded by the EIS curves are plotted in Supplementary Fig. 2. The ionic conductivity of the ionic solution decreases slightly with increasing FA content, probably because the viscosity increases with FA content. All ionic conductivities decline similarly as the temperature drops from 20 °C to 0 °C. However, below 0 °C, FA added at any proportion increases the ionic conductivity from that at FA0. In particular, FA50 retains is high ionic conductivity (5.90 $mS\,cm^{-1}$) down to $-45$ °C, whereas the ionic conductivity of FA0 dramatically drops to 0.78 $mS\,cm^{-1}$ at $-45$ °C.

The molecular interactions in the ionic solutions were investigated using Fourier transform infrared spectroscopy (FTIR) (Fig. 1c and Supplementary Fig. 3). The spectra of all ionic solutions exhibit a characteristic peak at 2256 $cm^{-1}$ corresponding to the $C \equiv N$ stretching vibrations of cyanide ions ($CN^-$), indicating that the FA additive did not affect the complex structure of $FeCN^{4-/3-}$. Because the O–H and N–H bands of $H_2O$ and FA, respectively, coincide at around 3400 $cm^{-1}$, increasing the FA content caused no obvious peak displacement but the peaks were visibly separated. Non-negligible blue shifts of the C = O stretching vibrations (amide I) of FA at 1630–1685 $cm^{-1}$, N–H bending vibrations (amide II) of FA at 1600–1640 $cm^{-1}$, and C–H stretching vibrations of FA at 2850–2890 $cm^{-1}$ are observed with increasing FA content. In contrast, the peak positions of the C–N stretching vibrations (amide III) of FA at 1350–1430 $cm^{-1}$ are identical in the spectra of all samples. The results reveal at least three different forms of hydrogen bonds between FA and water (Supplementary Fig. 4)[38]. Hence, the $H_2O$–$H_2O$ hydrogen bonds are easily broken by the stronger hydrogen bonds of FA–$H_2O$, substantially lowering the $T_f$ of the hybrid electrolyte and thereby improving its ionic conductivity at low temperatures.

The i-TE liquid cell with the hybrid electrolyte maintains its excellent thermoelectric performance near cryo temperatures. In this study, the $P_{max}$ was optimized by adjusting the FA content and the electrode separation gap $h$. In our previous work, $P_{max}$ was adopted as a performance indicator for seeking the maximum hot-side temperature ($T_{h,max}$) and minimum cold-side temperature ($T_{c,min}$)[18]. Herein, we

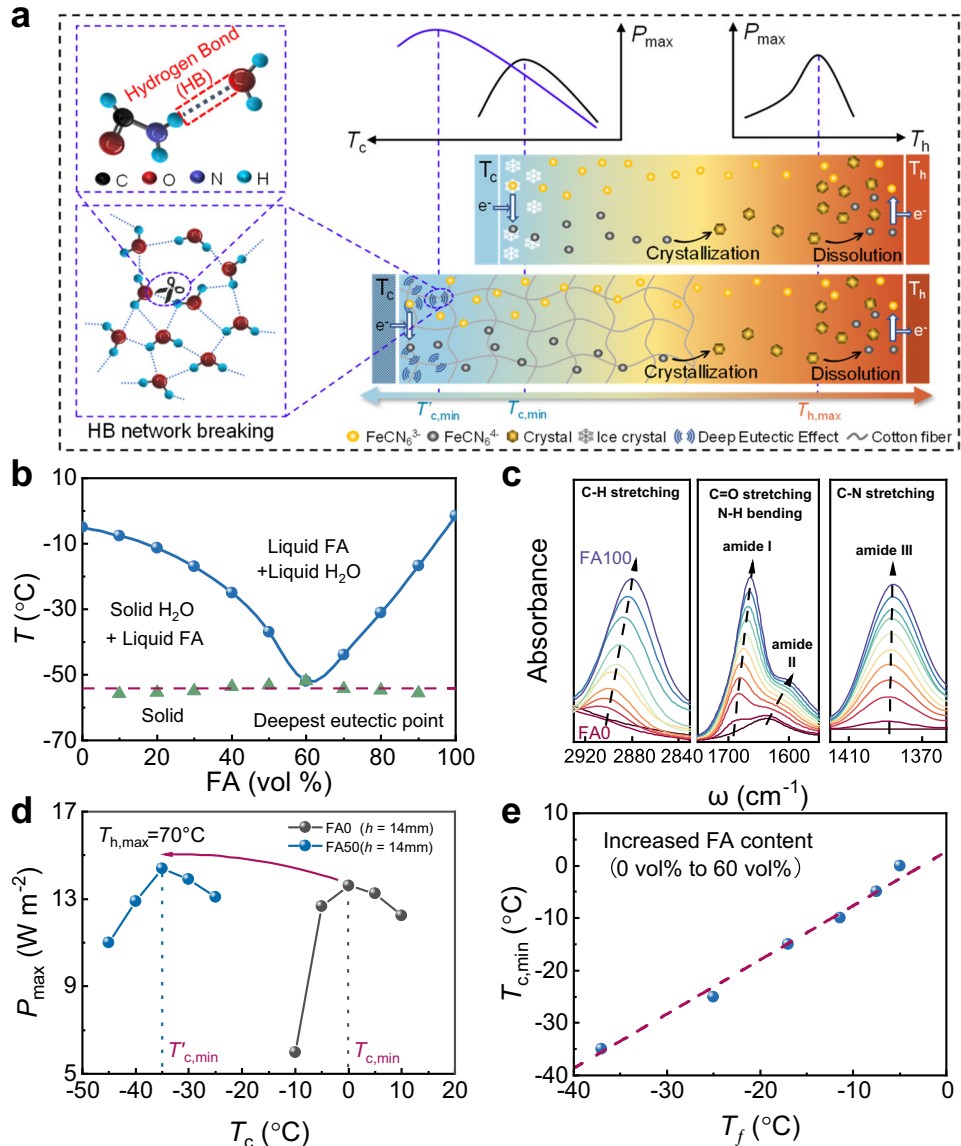

**Fig. 1 | Principle of anti-freezing. a** Design principal of the FA–H$_2$O/ GdmCl–FeCN$^{4-/3-}$ ionic thermoelectric (i-TE) liquid cell (FA = formamide, GdmCl = guanidinium). **b** phase diagram of the hybrid electrolyte. **c** FTIR spectra of the hybrid electrolytes (FA0, FA10, FA20–FA100). **d** Plot of maximum power $P_{max}$ of the FA50 ($h$ = 14 mm) and FA0 ($h$ = 14 mm) i-TE liquid cells versus cold-side temperature $T_c$ (with fixed hot-side temperature $T_h$ = 70 °C). **e** Plot of $T_f$ versus $T_{c,min}$ for H$_2$O/ FA–FeCN$^{4-/3-}$–GdmCl i-TE liquid cells with increasing FA content (blue points) and its corresponding linear fitting (red dashed line).

first discuss the effects of FA content on $T_{c,min}$ and $P_{max}$ in the H$_2$O/ FA–0.4 M FeCN$^{4-/3-}$–3 M GdmCl liquid cell. At a GdmCl concentration of 3 M, the i-TE liquid cell with the 0.4 M FeCN$^{4-/3-}$ complex produces the highest thermopower (Supplementary Fig. 5). The theoretical upper limit of $T_{h,max}$ in this system is 100 °C. At higher temperatures, the system fails due to boiling of the electrolyte. Considering the thermal stability of the thermal-sensitive crystals in this system, the $T_{h,max}$ was set at 70 °C. Meanwhile, the $T_{c,min}$ decreases with increasing FA content. More specifically, at $x$ = 0, 10, 20, 30, 40, 50, and 60 vol.%, the $T_{c,min}$ values were 0 °C, −5 °C, −10 °C, −15 °C, −25 °C, −35 °C, and −50 °C, respectively, corresponding to $\Delta T_{max}$ increases of 70 °C, 75 °C, 80 °C, 85 °C, 95 °C, 105 °C, and 120 °C, respectively (Supplementary Fig. 6 and Supplementary Fig. 7). Supplementary Fig. 8 shows the $V_{oc}$ and $I_{sc}$ of the i-TE liquid cell ($h$ = 14 mm) working at their corresponding $\Delta T_{max}$ values. The $V_{oc}$ increases with increasing FA content (245, 255, 264, 288, and 310 mV at $x$ = 10, 20, 30, 40, and 50 vol.%, respectively). The increase from FA0 to FA10 is slight. The $I_{sc}$ trends oppositely because the increase

in viscosity and decrease in conductivity with increasing FA content, along with the decrease in service temperature. However, the $V_{oc}$ and $I_{sc}$ of FA60 show a marked decline, probably reflecting the decreased thermopower of this cell. As shown in Fig. 1d, the anti-freezing performance and $P_{max}$ of the optimized i-TE liquid cell (FA50 with $h$ = 14 mm) are largely improved from those of FA0. After manipulating the solvent molecules, the $T_{c,min}$ decreased from 0 °C to −35 °C°, suggesting excellent frost resistance. The $P_{max}$ of FA50 ($h$ = 14 mm) reached 14.4 W m$^{-2}$ when $T_c$ = −35 °C and $T_h$ = 70 °C and the corresponding $\Delta T_{max}$ was 150% higher than that of FA0 ($h$ = 14 mm) with the low-entropy aqueous electrolyte ($P_{max}$ = 13.5 W m$^{-2}$, $T_{c,min}$ = 0 °C). Note that the $T_{c,min}$ of FA60 was further decreased to −50 °C, the lowest among the FA series (FA0 to FA100; see Supplementary Fig. 9), but the $P_{max}$ of FA60 was reduced by the low thermopower. To clarify the relationship between $T_{c,min}$ and the eutectic behavior of the electrolyte, Fig. 1e plots $T_f$ versus $T_{c,min}$ for the i-TE liquid cells. The plot is almost linear with an slope of ~1, indicating that the $T_{c,min}$ obtained in the performance test well

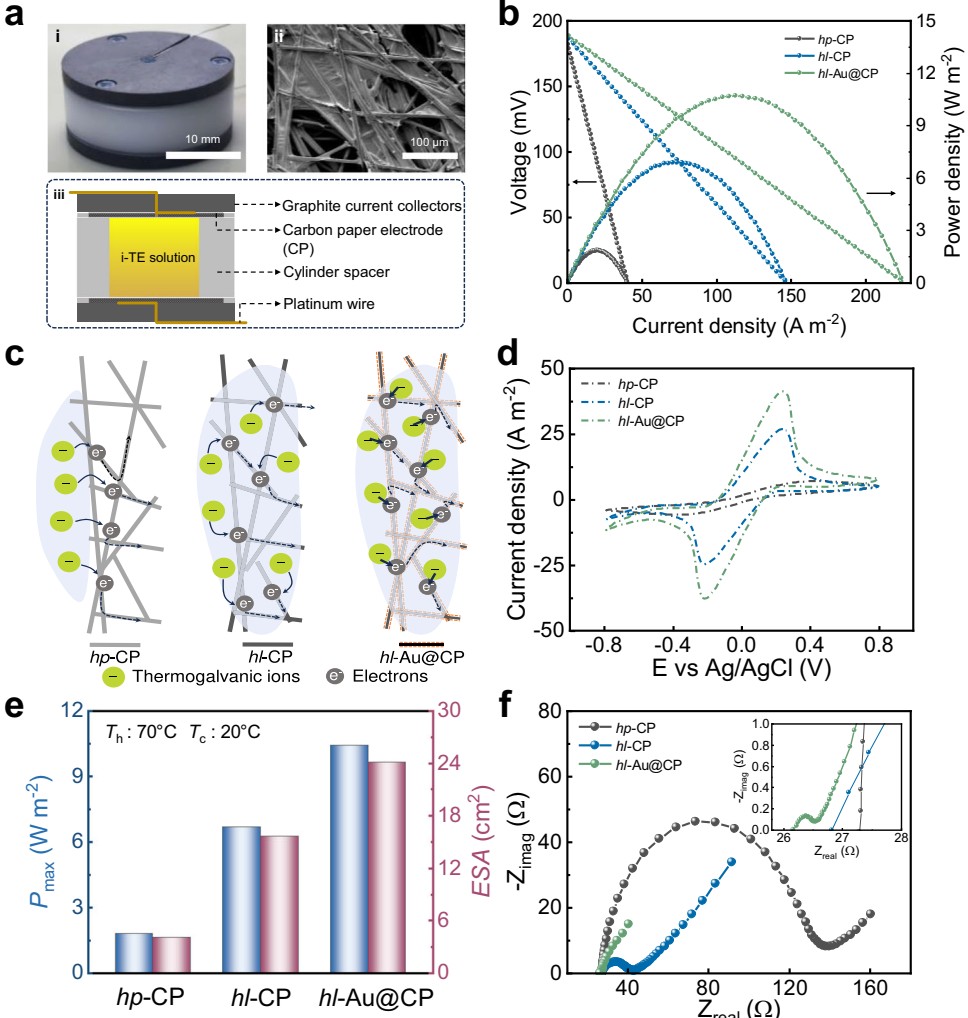

**Fig. 2 | Electrode optimization of the FA–H$_2$O/GdmCl–FeCN$^{4-/3-}$ i-TE liquid cell.** **a** Components of the cell. i: Entity profile of the cell. ii: Scanning electron microscope (SEM) image of a typical carbon paper (CP) electrode. iii: Structure diagram of the cell. **b** Voltage and output power density versus current density of the i-TE liquid cell (FA0, $h = 14$ mm) with three different electrodes (hydrophobic carbon paper (CP), hydrophilic CP, hydrophilic Au@CP) with $T_h$ and $T_c$ set to 70 °C and 20 °C, respectively. **c** Schematic showing the effect of hydrophilic treatment and a gold-nanoparticle coating on the structure and properties of the CP electrode. **d** Cyclic voltammograms of the hydrophobic CP, hydrophilic CP, and hydrophilic Au@CP electrode. **e** Maximum output power density $P_{max}$ and electroactive surface area (ESA) values of i-TE liquid cells (FA0, $h = 14$ mm) with different electrodes. **f** Nyquist plots of the hydrophobic CP, hydrophilic CP, and hydrophilic Au@CP electrode (inset is a magnified plot of the linear regions).

corresponds to the eutectic behavior of the H$_2$O/ FA–Fe(CN)$_6$$^{4-/3-}$–GdmCl electrolyte.

## Electrode structure and current density

The electrode structure plays a critical role in the energy output of an i-TE liquid cell[12,17]. A rough electrode with a high specific surface area improves the short-circuit current density ($I_{sc}$) of the cell[17]. Here, the i-TE liquid devices were assembled from two carbon paper (CP) electrodes, the electrolyte, and a thermal insulation separator. They were encapsulated within two graphite current collectors and a cylinder spacer that defines the electrode separation gap ($h$) (Fig. 2a and Supplementary Fig. 10). The measured thermopowers of the 0.4 M K$_3$Fe(CN)$_6$/K$_4$Fe(CN)$_6$ and 0.4 M K$_3$Fe(CN)$_6$/K$_4$Fe(CN)$_6$–3 M GdmCl cells with this structure were 1.4 and 3.7 mV K$^{-1}$ respectively (Supplementary Fig. 11), consistent with their reported values[22]. The FA additive slightly decreases the thermopower of the H$_2$O/$x$ FA–$y$ FeCN$^{4-/3-}$–$z$ GdmCl solution. For example, the thermopower of FA50 solution is 3.1 mV K$^{-1}$, 16% lower than that of FA0 solution (Supplementary Fig. 12). FA weakens the chaotropic-chaotropic interaction between GdmCl and [FeCN$_6$]$^{4-}$, which lowers the

reaction entropy of the redox couples and thus the thermopower. The UV-Vis spectra of FA-free K$_4$Fe(CN)$_6$ with and without GdmCl (Figure S13a), and K$_4$Fe(CN)$_6$-GdmCl with varying FA (Figure S13b) were obtained to characterize the GdmCl-[FeCN$_6$]$^{4-}$ interaction. First, the [FeCN$_6$]$^{4-}$ peak shifts from 178 nm to 209 nm upon GdmCl addition, indicating the chaotropic-chaotropic interaction between GdmCl and [FeCN$_6$]$^{4-}$. However, in Figure S13b, the [FeCN$_6$]$^{4-}$ peak of the i-TE solution of K$_4$Fe(CN)$_6$-GdmCl (FA/Water) reverses from 209 nm to 184 nm, indicating that the chaotropic-chaotropic interactions are weakened by FA. Moreover, we compared the thermopower of two i-TE solution series, namely 0.4 M FeCN$^{4-/3-}$ and 0.4 M FeCN$^{4-/3-}$–3M GdmCl with different FA (Figure S14). It shows that, without GdmCl, the thermopower is nearly constant with FA. In contrast, in 0.4 M FeCN$^{4-/3-}$–3M GdmCl, the thermopower decreases with increasing FA. This suggests that FA affects the hydrate structure of GdmCl rather than [FeCN$_6$]$^{4-}$, and consequently the GdmCl-[FeCN$_6$]$^{4-}$ interaction.

CP electrodes are known to possess excellent electrical conductivity and a self-supporting carbon-fiber network structure that provides an abundant electrochemically active interface for electron

exchange between redox couples[39]. Here, we noted that hydrophilicity of the electrode is another important determiner of high $I_{sc}$. The optical contact angle of the as-received CP is 114.6° (Supplementary Fig. 15). The as-fabricated i-TE solutions cannot diffuse into the CP electrode to form a heterogeneous interface. After treatment with oxygen plasma, the i-TE solution droplets completely infiltrated and penetrated the electrode interior, suggesting good hydrophilic performance. Hereafter, the as-received CP is referenced as hydrophobic CP (hp-CP) while the hydrophilically treated CPs are called hydrophilic CPs (hl-CP). As shown in Fig. 2b, the $P_{max}$ is 3.7 times higher in the hl-CP | FA0|hl-CP i-TE liquid cell (h = 14 mm) than in the hp-CP liquid cell when $T_h$ and $T_c$ are set to 70 °C and 20 °C, respectively.

Next, a 40-nm-thick Au layer was installed to reduce the electrode-interface resistance relative to electron exchange between the $K_3Fe(CN)_6/K_4Fe(CN)_6$ redox couple. The porous structure of the as-received CP (pore size 10–100 μm) was unaffected by the Au coating layer. For simplicity, this hydrophilic CP electrode with the 40-nm-thick Au coating is referenced as hl-Au@CP. The $P_{max}$ of the hl-Au@CP | FA0|hl-Au@CP i-TE liquid cell (h = 14 mm) was further improved to 10.43 W m$^{-2}$, 5.8 times higher than $P_{max}$ of the hp-CP cell (Fig. 2b). The schematic in Fig. 2c shows the effects of the hydrophilic treatment and gold-nanoparticle coating on the structure and properties of the CP electrode. The hl-CP electrode provides more electrochemically active sites than hp-CP because the surface energy between the electrode surface and electrolyte is higher for hl-CP than for hp-CP, promoting the infiltration of thermogalvanic ions into the internal microstructure. The gold nanoparticles on hl-Au@CP effectively reduce the electrode-interface resistance and accelerate the electron transfer process of the redox reaction[40]. The electroactive surface area (ESA) can be determined from the peak current density ($I_p$) in the cyclic voltammograms (CVs) of the electrodes (Fig. 2d). According to the Randles−Sevcik equation, a high faradaic peak current of an electrode exhibiting reversible kinetics indicates a high ESA[27]. The ESAs of the hp-CP, hl-CP, and hl-Au@CP electrodes were estimated as 4.11, 15.69, and 24.1 cm$^2$, respectively. Note that the ESA values indicate only the performances of the CP electrodes, not their actual areas. In general, a higher ESA corresponds to a higher $P_{max}$ (Fig. 2e). Furthermore, the charge transfer resistances ($R_{ct}$, diameter of the semicircle in a Nyquist plot) of the hp-CP, hl-CP, and hl-Au@CP electrodes were 102, 16, and 0.96 Ω, respectively. The diffusion-controlled impedance (Warburg impedance $Z_w$, reflected in the part of the Nyquist plot with a near-45° slope in the medium−high frequency domain)[41] was much lower in the hl-Au@CP electrode (5.62 Ω) than in the hp-CP (20.3 Ω) and hl-CP (15.4 Ω) electrodes, indicating fast electron transfer and ion diffusion within the hl-Au@CP electrode (Fig. 2f).

## Structure and effective temperature difference of the i-TE liquid cell
The electricity generation performance of i-TE cells relies on the i-TE materials and the effective temperature difference ΔT. Convection in the i-TE liquid cell enables continuous operation of the redox couple[42] but lowers the thermal resistance and reduces the ΔT[28]. Consequently, the electrode separation gap and thermally insulating separator (TIS) are key components of the structural design of an i-TE liquid cell. For instance, by using a hydrophilic cellulose sponge, Baughman et al. improved the ΔT of the cell from 63 to 95 °C[28]. Here, a cotton-fiber TIS was immersed in the i-TE solution to enlarge the thermal resistance of the i-TE liquid cell (Supplementary Fig. 16). Figure 3a plots the effective ΔT across the i-TE liquid cells with and without the TIS as functions of heat flux. The tested cells were $H_2O/FA−0.4$ M FeCN$^{4−/3−}$, $H_2O/FA−0.4$ M FeCN$^{4−/3−}$ −3 M GdmCl (FA50), and FA50 with TIS. First, one observes that GdmCl enlarges the ΔT by reducing the convection. Second, the TIS effectively suppresses heat convection and increases the thermal resistance of the cell (The thermal conductivity of the electrolyte and electrode are shown in Supplementary Table 1). The TIS raised the

maximum ΔT of the FA50 cell by 244% from that of the FA50 cell without the TIS (100 °C versus 41 °C under a heat flux of 13 W cm$^{-2}$). Furthermore, the TIS desensitized the i-TE liquid cell to the temperature-gradient direction (Fig. 3b), which affects the power generation of the cell. Typically, placing the cold electrode above the hot electrode achieves a higher $P_{max}$ than placing the hot electrode above the cold electrode or placing vertical cold and hot electrodes. In the former configuration, incessant convection mixing homogenizes the electrolyte and increases its ionic conductivity[43]. In this study, the TIS suppressed convection through the electrolyte and fixed thermo-sensitive crystal precipitation, so the $P_{max}$ was insensitive to the temperature-gradient direction.

Figure 3c shows the effect of electrode separation gap (h) on the $P_{max}$ of the hl-Au@CP | FA50-TIS|hl-Au@CP cell working at its corresponding $ΔT_{max}$. Decreasing h from 14 to 8 mm did not affect the $T_{c,min}$, indicating that $T_{c,min}$ is an inherent property of the electrolyte and independent of device structure (Supplementary Fig. 17). $V_{oc}$ was also unchanged, indicating that the thermopower is an intrinsic property of the cell (Supplementary Fig. 18). The $I_{sc}$ initially increased from 185 to 208 A m$^{-2}$ as h decreased from 14 to 12 mm, reached a maximum of 226 A m$^{-2}$ at h = 10 mm, and finally decreased to 214 A m$^{-2}$ at h = 8 mm. Consequently, a $T_{c,min}$ of −35 °C and a $P_{max}$ of 17.5 W m$^{-2}$ were obtained at h = 10 mm. The shorter of the electrode separation gap would have less internal resistance, hence resulting in the increase $I_{sc}$. In order to clarify the abnormal decreased $I_{sc}$ as the h further decreased to 8 mm, we have conducted simulation with COMSOL Multiphysics 6.0 to investigate the effect of electrode separation gap on the thermo-gavanic convection. In the i-TE solution of $K_3Fe(CN)_6/K_4Fe(CN)_6$, the ionic convection is due to the accumulated $Fe(CN)_6^{3-}$ from oxidation of $Fe(CN)_6^{4-}$ at hot side, resulting into back flux of $Fe(CN)_6^{3-}$ from hot side to cold, and vice versa for cold side[19]. The convection of redox couples also afford the i-TE cells work continuous, and also have higher energy output than that with only thermodiffusion ions. Figure 3d compares the simulated convection of the i-TE cell of $H_2O/FA−0.4$ M FeCN$^{4−/3−}$ −3 M GdmCl (FA50) with different electrode gap, and a fixed temperature difference of 105 °C ($T_h = 70$ °C, $T_c = −35$ °C). It clearly shows that the electrode gap has a significant effect on the convection speed, showing that the redox pair circulation slow down as the electrode separation gap decreases. This reduced convection would result in a decreased $I_{sc}$. In other words, the decreased h has two opposite effects on the $I_{sc}$. The balance between the reduced internal resistance and suppressed convection results in the optimized h. Additionally, we also noted that the convection of redox couple also resulted in the non-uniform temperature gradient both vertically and horizontally (Supplementary Fig. 19), which add the complex to understand the contribution of the microscopic ionic behavior to macroscopic thermovoltage. To identify the $T_{c,min}$, $I_{sc}$, $P_{max}$, and effective resistance values ($R_{eff}$) of the optimal i-TE liquid cell, namely, hl-Au@CP | FA50-TIS|hl-Au@CP (h = 10 mm), the $I_{sc}$, $P_{max}$, and $R_{eff}$ are plotted as functions of $T_c$ in Fig. 3e and Supplementary Fig. 20. The $I_{sc}$ gradually decreased from 233 A m$^{-2}$ at $T_c = −25$ °C to 227 A m$^{-2}$ at $T_c = −35$ °C and suddenly decreased to 203 A m$^{-2}$ at $T_c = −40$ °C and to 177 A m$^{-2}$ at $T_c = −45$ °C. The dramatic change in the $I_{sc}$ trend at $T_c = −35$ °C can be understood from the $R_{eff}$ plot. As $T_c$ decreased from −25 °C to −30 °C to −35 °C, $R_{eff}$ increased from 4.6 to 4.9 to 5.2 Ω because the ion transport rate reduces with decreasing temperature. However, a phase transition in the electrolyte increased the $R_{eff}$ to 6 Ω at $T_c = −40$ °C and 7 Ω at $T_c = −45$ °C. An obvious turning point at −35 °C also appears in the growth curve of $V_{oc}$ from −25 °C to −45 °C (Supplementary Fig. 21). From the $P_{max}$, $T_{c,min}$ was therefore determined as −35 °C. Figure 3f compares the obtained $P_{max}$ and $T_{c,min}$ in this work with those of other reported i-TE liquid cells[12,13,15,22–24,26,28,30,43–45]. To our knowledge, we have achieved the highest power density near cryo temperatures and the lowest cold-side temperature. Improving the anti-freezing solvent environment and accelerating the dynamic ion migration was

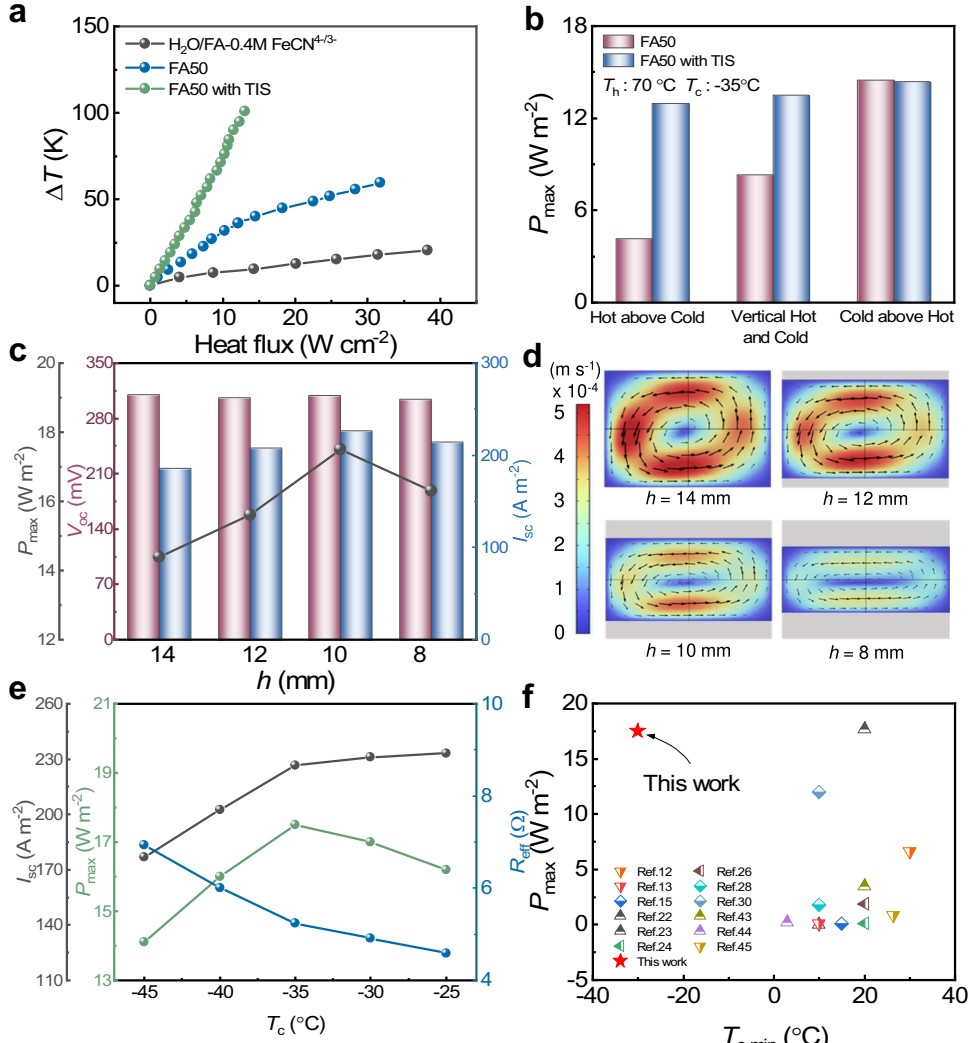

**Fig. 3 | Structural optimization of the H₂O/FA–FeCN⁴⁻/³⁻–GdmCl i-TE liquid cell.**
**a** Temperature differences between the cold and hot electrodes as functions of heat flux in three electrolyte configurations (FA50 with neither GdmCl nor a thermal insulation separator (TIS), FA50 with GdmCl but without TIS, FA50 with both GdmCl and TIS). **b** $P_{max}$ of the i-TE liquid cell in three electrode orientations with or without the TIS. **c** Open circuit voltage $V_{oc}$, short-circuit current $I_{sc}$, and $P_{max}$ in the H₂O/FA–FeCN⁴⁻/³⁻–GdmCl i-TE liquid cell with different electrode separations gaps. **d** Simulated convection profiles of the FA50-TIS i-TE liquid cell with varying electrode separations gaps at $T_h = 70\,°C$ and $T_c = -35\,°C$. **e** $I_{sc}$, $P_{max}$, and effective resistance $R_{eff}$ of the FA50 cell ($h = 10\,mm$) versus $T_c$. **f** Comparison of $P_{max}$ and $T_{c,min}$ between the proposed and reported i-TE liquid cells[12,13,15,22–24,26,28,30,43–45].

considered to stabilize the mass transfer for redox couples at a lower $T_{c,min}$, thereby increasing the $P_{max}$.

## Long-term power generation

Next, we investigated the energy output performance and cyclic performance of the as-fabricated $hl$-Au@CP | FA50-TIS|$hl$-Au@CP i-TE liquid cell with the optimized separation gap of the electrodes ($h = 10\,mm$). The i-TE generator mode[17] consists of three stages: voltage build-up, power output, and reactivation. During the first stage, an electric field is generated by the thermogalvanic transfer of FeCN⁴⁻/³⁻ from the cold to the hot electrode. During the second stage, an external load is connected and electrons flow from the hot to the cold electrode, decreasing the internal electrostatic field and voltage. During the final stage, the consumed species at the electrodes are replenished by diffusing FeCN⁴⁻/³⁻, canceling the temperature gradient and recovering the voltage. Figure 4a presents the working voltage and current curves of the optimized i-TE liquid cell with a 50 Ω external resistance. The maximum $\Delta T$ of the i-TE liquid cell was 105 K ($T_{c,min} = -35\,°C$, $T_{h,max} = 70\,°C$). The $V_{oc}$ initially rose to 310 mV and then dropped to 195 mV after 2 h of power output to the external

circuit. Finally, the $V_{oc}$ reached 0 mV due to short circuiting and $\Delta T$ removal. In Stage (ii), the current reduced from 6.2 to 3.9 mA after 2 h of power output. The output power density during this stage was measured with various external resistors (10–90 Ω; see Fig. 4b). Figure 4c plots the energy density calculated by integrating the power-output curves for 2 h as a function of resistance. Owing to the extended $\Delta T_{max}$ and optimized electrode and cell structure, the as-fabricated i-TE liquid cell achieved a two-hour energy density ($E_{2h}$) of 27 kJ m⁻² with a 50 Ω external resistor even at near cryo temperatures. This performance far exceeds those of other reported quasi-solid-state i-TE cells (Fig. 4d)[16–18,46–51]. The $E_{2h}$ of FA0 ($T_c$: 0 °C) and FA50 ($T_c$: −35 °C) was also compared (Supplementary Fig. 22). The matching resistance with the highest $E_{2h}$ of FA0 is smaller than that of FA50, indicating the lower internal resistance of FA0 ($T_c$: 0 °C). However, the smaller working temperature range of FA0 results in a lower working voltage and a lower $E_{2h}$, which compromise its output performance.

The cyclic working–resting performance of the as-fabricated i-TE liquid cell was measured in generator mode during one week of continuous cycling with $T_c = -35\,°C$ and $T_h = 70\,°C$. During each one-day cycle, the working–rest mode exhibited the above-mentioned voltage

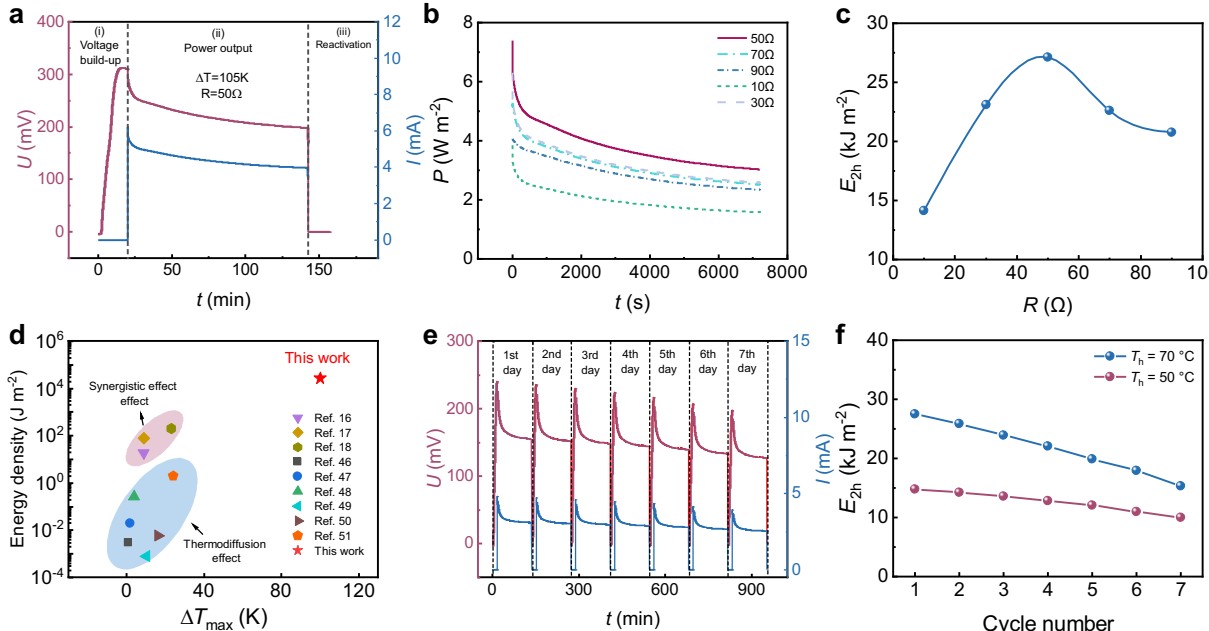

**Fig. 4 | Long-term power generation of the *hl*-Au@CP | FA50-TIS|*hl*-Au@CP i-TE liquid cell (*h* = 10 mm) at $T_c$ = −35 °C in i-TE generator working mode.**
**a** Measured voltage and current curves during three working stages ($T_h$ = 70 °C).
**b** Output power density measured over 2 h in stage (ii) using different external resistors. **c** Corresponding generated energy density as a function of external resistors, calculated by integrating the output power over time (2 h) shown in (**a**).

**d** Performance comparison of energy density between the proposed i-TE cell and reported i-TE cells based on thermodiffusion or synergistic effects[16–18,46–51].
**e** Measured voltage and current curves during one week of continuous cycling ($T_h$ = 50 °C). **f** Decay of two-hour energy output $E_{2h}$ of the i-TE liquid cell during one week of cyclic service at $T_h$ = 70 °C and 50 °C.

build-up (~20 min), power output with a 50 Ω resistor (2 h), and self-reactivation with short circuiting at $\Delta T$ = 0 K (4 h). The cell then rested (~18 h) until the following day (The time of reactivation was set to 4 hours to ensure that the open-circuit voltage returned to zero). The measured voltage and current curves of the build-up voltage and power-output stages during the first seven days and the output power density during the second stage on the first, third, fifth, and seventh days are shown in Supplementary Fig. 23. The 2 h energy density ($E_{2h}$) decreased from 27 kJ m$^{-2}$ on day 1 to 15 kJ m$^{-2}$ on day 7, for which the following reasons may account: first, the thermosensitive crystallization dissolution kinetics were presumably slowed by long cycle service at $T_h$ = 70 °C, resulting in the insufficient supply of active ions required for the electrode reaction. Second, the formation and dissolution process of thermosensitive crystallization has a long-term impact on the electrode, such as the electrode surface is gradually covered with a passivation film (Supplementary Fig. 24) which hinders the further diffusion and reaction of active particles on the electrode. To verify this conjecture, we compared the $E_{2h}$ decays of the i-TE liquid cell during one week of cyclic service at $T_h$ = 70 °C and 50 °C (Fig. 4e, f). The cyclic performance of the i-TE liquid cell was higher during service at $T_h$ = 50 °C than during service at $T_h$ = 70 °C. At $T_h$ = 50 °C, the $E_{2h}$ was 14.8 kJ m$^{-2}$ in the first cycle and decreased to 10 kJ m$^{-2}$ in the seventh cycle, demonstrating a good retention rate of 68%. The specific composition of the passivation layer covering the electrode was also characterized with XRD (Supplementary Fig. 25a) and XPS (Supplementary Fig. 25b) measurement. It can be observed that the passivation layer is essentially the thermosensitive crystal formed by the combination of GdmCl and Fe(CN)$_6$$^{4+}$. During the cyclic testing process, the grain size gradually decreased and easily adhered to the electrode surface to form the passivation layer, resulting in the performance degradation. The cyclic working–resting performance of FA0 ($T_h$ = 70 °C, $T_c$ = 0 °C) was also conducted (Supplementary Fig. 26). The decay of two-hour energy output $E_{2h}$ of FA50 and FA0 during one week of cyclic service were compared (Supplementary Fig. 26c). Despite exhibiting more severe degradation than FA0, FA50

still had a higher total power capacity of 14 hours (152.4 kJ m$^{-2}$ for FA50 compared to 142.2 kJ m$^{-2}$ for FA0). The instantaneous output power ($P_{max}$) of the cell was affected by the long-term cyclic service, and the $P_{max}$-$T_c$ graph of FA0 and FA50 after the one-weak cycle test was given by Supplementary Fig. 27. The $P_{max}$ of both exhibit varying degrees of attenuation (9.97 W m$^{-2}$ for FA50 and 10.2 W m$^{-2}$ for FA0). The lower $P_{max}$ of FA50 was attributed to its poorer cycling performance, which was influenced by factors such as larger working temperature difference, lower cold-end temperature, lower conductivity and higher viscosity of FA50 solution.

## 25-cell i-TE module
Finally, the optimized i-TE liquid cell, namely, *hl*-Au@CP | FA50−TIS|*hl*-Au@CP (*h* = 10 mm), was assembled into a 25-cell i-TE module (Fig. 5a). The $V_{oc}$, $I_{sc}$, and $P_{max}$ produced by the module were 6.9 V, 68 mA, and 131 mW, respectively, with $\Delta T$ = 105 K ($T_c$ = −35 °C and $T_h$ = 70 °C) (Fig. 5b). Owing to its considerable output power, the module could directly drive a light-emitting diode (LED) array (Fig. 5c). This test demonstrates that our device can power electronics near cryo temperatures, confirming its practical utility. Figure 5d compares the voltage and power of our as-fabricated i-TE liquid cell module devices with those of other reported i-TE and e-TE devices[15–18,22,25,27,52,53]. Our module doubled the voltage from those of the reported devices based on other i-TE systems. Our work extends the application scenarios of i-TE liquid cells to the extremely cold regime, potentially realizing self-powered supplies that harvest waste heat for IoT sensors in extremely cold environments. Our work also promises scientific activity in frigid polar regions.

## Methods
### Materials
K$_3$Fe(CN)$_6$ ($M_W$ = 329.25, ≥99.5%), K$_4$Fe(CN)$_6$ 3H$_2$O ($M_W$ = 422.39, 99.0%), guanidine hydrochloride (GdmCl) ($M_W$ = 95.53, 99.0%), and formamide (FA) ($M_W$ = 45.04, 99.0%) were purchased from Aladdin Industrial Corporation (Shanghai, China). Potassium chloride (KCl)

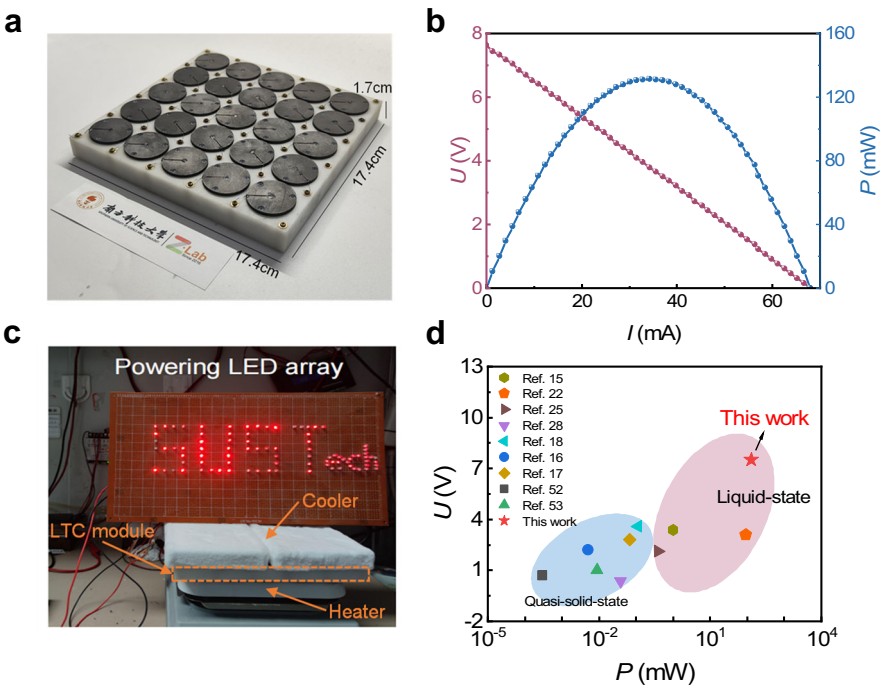

**Fig. 5 | i-TE liquid cell module composed of 25 units in series. a** Schematic diagram of the module. **b** Output voltage and power versus output current of the module device. **c** Powering an LED array with no additional voltage boosters.

**d** Performance comparisons of output voltage and power between the proposed and previously reported i-TE devices[15–18,22,25,27,52,53].

($M_W$ = 74.55, 99.8%) was provided by Macklin Biochemical Co., Ltd. (Shanghai, China). All chemical reagents were used as received without further purification. Graphite plates and CP were purchased from JingLong Special Carbon Company, Ltd. (China) and Toray Industries, Inc. (Japan), respectively.

**Mechanism characterization**

The freezing points of the $H_2O$/FA–$FeCN^{4-/3-}$–GdmCl samples were characterized using a differential scanning calorimeter (Mettler Toledo DSC1, USA). The ionic conductivities of the samples were calculated from the first intercept of the high-frequency semicircle in the results of EIS measurements at different temperatures, where the response is fully resistive. The structural and chemical bonding characteristics of the electrolyte were examined using FTIR-attenuated total reflectance techniques (Bruker Vertex 70 v, province, country). The electrochemical performances were measured on an electrochemical workstation (Zennium Pro, Germany) with a three-electrode configuration. A CP piece was used as the working electrode and platinum and Ag/AgCl electrodes were used as the counter and reference electrodes, respectively. The CV scanning was performed at 10 mV s$^{-1}$. The Randles–Sevcik equation is given by $I_p = 0.4463nFAC\sqrt{\frac{nFvD}{RT}}$, where $I_p$ is the faradaic peak current, $n$ is the number of electrons transferred during the redox reaction, $F$ is Faraday's constant, $A$ is the ESA, $C$ is the concentration of the probe molecule, $v$ is the potential scan rate, $D$ is the diffusion coefficient, $R$ is the universal gas constant, and $T$ is the temperature. EIS measurements were conducted between 10 kHz and 50 mHz with an AC amplitude of 10 mV. Both CV and EIS were run using 0.01 M ferro/ferricyanide solution with 0.1 M KCl as the supporting electrolyte in aqueous media to reduce the ohmic overpotential in the cell.

**Fabrication of the i-TE liquid cell**

The i-TE liquid cell was assembled with a laminar structure of two graphite current collectors, two CP electrodes, a thermal separator attached to the cold-side electrode, a cylinder spacer composed of polyformaldehyde (commercial sources), the electrolyte volume at the

cell center, and two rubber O-rings (Supplementary Fig. 10). The spacer defines the electrode separation gap $h$. To prepare the hydrophilic Au@CP electrode, the CP was first treated with oxygen plasma (SUNJUNE PLASMA VP-R5, China) for 10 minutes to induce strong hydrophilicity. Next, a 40-nm-thick Au coating was deposited on the carbon fibers for 120 s at a current of 30 mA using an MC1000 ion sputter (Hitachi Ltd, Japan). A cotton-fiber laminating thermal separator was fabricated by compressing commercial cotton balls with approximate diameters of 20 μm (Supplementary Fig. 16). To form the FA50 electrolyte (as an example; the other electrolytes were prepared similarly), $K_3Fe(CN)_6$ (1.98 g) and $K_4Fe(CN)_6$ 3$H_2O$ (2.56 g) were dissolved in 7.5 mL deionized (DI) water. After adding 4.34 g of GdmCl, the solution was stirred for 10 min, yielding a light-yellow solution with precipitation. Finally, 7.5 mL FA was added and the solution was stirred for 5 min to obtain the optimized electrolyte.

**Performance measurements of the i-TE liquid cell**

The performance of the i-TE liquid cell was tested in an actual measurement setup. The temperature was controlled with a digital temperature control module (YEXIAN TCM-M207, China) and the voltage and temperature data were acquired using the LabVIEW program. The voltage was measured with a Keithley-2000 instrument and the current–voltage characterization of the device was performed with a Keithley 2400 instrument. There are approximately 100 points between 0 V to open-circuit voltage. The voltage sweep rate is 0.1 s per point. The voltage and current were measured by a Keithley-2400 and Keithley-6450, respectively, during the voltage build-up and power output stages of continuous power output. The thermal conductivity of the electrode was measured with hot-disk method, and the effective thermal conductivity of the electrolyte was measured with the steady-state method[15].

**Module preparation**

The module containing 25 integrated units consisted of a polyformaldehyde frame, graphite current collectors, CP electrodes, electrolytes, and titanium (Ti) wires. The frame size was 160 mm ×

160 mm × 15 mm (length × width × height) and contained 25 cells. Twenty-five pairs of graphite sheets were fixed on the frame by screws to prevent leakage while the CP electrodes were fixed between the graphite sheet and the frame. The cells were connected in series by Ti wires.

## Data availability

The source data used in this study are available in the Figshare database (https://doi.org/10.6084/m9.figshare.24968556). Extra data are available from the corresponding author (liuws@sustech.edu.cn) upon reasonable request.

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

## Acknowledgements
This work was supported by the Shenzhen Innovation Program for Distinguished Young Scholars (Grant Nos. RCJC20210706091949018), the Guangdong Innovative and Entrepreneurial Research Team Program (Grant Nos. 2016ZT06G587), and a Shenzhen Science Technology Fund (Grant Nos. KYDPT20181011104007). The authors acknowledge support from the Centers for Mechanical Engineering Research and Education at MIT and SUSTech. W.S.L acknowledges support from the Tencent Foundation through the XPLORER Prize.

## Author contributions
Shuaihua Wang and Yuchen Li conceived this project. Shuaihua Wang prepared the samples, and fabricated the cells. Shuaihua Wang and Mao Yu conducted experiment. Shuaihua Wang, Yuchen Li, Mao Yu and Qikai Li analyzed the data. Shuaihua Wang and Yupeng Wang performed COMSOL simulation. Shuaihua Wang prepared the manuscript. Huan Li, Jiajia Zhang, Yuchen Li and Weishu Liu revised the manuscript. Weishu Liu funding acquisition. Weishu Liu, supervision. All authors participated in the analysis and discussion.

## Competing interests
The authors declare no competing interests.
