## [Peer Review File · Nature Communications]

REVIEWER COMMENTS

Reviewer #1 (Remarks to the Author):

In this work, the authors added formamide and guanidinium into an aqueous electrolyte to broaden the working temperature range and enhance thermopower. Au-coated carbon electrodes and a thermally insulating separator were employed to improve the electricity generation performance of the thermoelectric cell. There are some issues to be addressed before considering acceptance.

(1) The authors need to explain the difference from the previously reported cryo-thermocell in terms of eutectic solvent strategy. (<https://doi.org/10.1002/adma.201901403>)

(2) The authors need to explain the difference from the previous paper that utilized thermally insulating separators from the standpoint of the thermal resistance effect. (<https://doi.org/10.1002/adma.201605652>)

(3) Could you explain the difference from the previous paper using plasma-treated carbon electrodes regarding electrode wettability? (<https://doi.org/10.1002/adma.201303295>)

(4) Could you provide the Pmax-Tc graphs (Figure 1d) of FA0 and FA50 after the cycle test shown in Figure 4f?

(5) Please compare the E2h of FA0 (Tc: 0 °C) and FA50 (Tc: -35 °C).

(6) Why does the FA additive reduce the thermopower?

(7) What is the voltage sweep rate when measuring performance? (e.g., Fig. 1d) Does it affect the performance?

(8) Please provide the voltage and output power density versus current density for Figure 3d in the supporting information.

(9) It is encouraged to add Fig. 3a and b for FA50 results.

(10) Does a FA added electrolyte have a constant thermopower over a range of temperatures (Tc,min~Th)?

Reviewer #2 (Remarks to the Author):

1. In the introduction part, the eutectic solvent strategy should be further introduced to highlight their advantages in the possible application of cryo-temperature region.

2.As demonstrated, the wettability of CP electrode by electrolyte plays a significant role to enhance the thermoelectrochemical performance such as power density. However, Figure S13 only provides the contact angle of hydrophobic CP (hp-CP) and hydrophilic CP (hl-CP). In my view, the contact angle of used hl-Au@CP also should be provided.

3.To my knowledge, the thermal conductivity of electrolyte and electrode affect the performances of i-TEs. Please supplement some related results.

4.As shown in Figure 4f, why does FA50 i-TE exhibit a voltage decay with the one-week continuous cycling? By the way, in the caption of Figure 4f, the first five days should be seven days? Same question also involved in the related Figure S19.

5.The morphology of electrodes after cycling test should be provided.

6.Some format type in the references should be corrected, such as Ref. 3, 21, 36.

Response to reviewers

Nature Communication

Manuscript ID: NCOMMS-23-44213-T

Title: High-Performance Cryo-Temperature Ionic Thermoelectric Liquid Cell
Developed through a Eutectic Solvent Strategy

Dear Reviewers:

We would like to thank you for your kindness in offering us the opportunity to improve our manuscript as per the valuable comments, which provided important insights to improve the quality of this work. Herein, we provide a detailed point-by-point response to the comments. In the following passages, comments are in *italics*, followed by our response in black. Revisions in response to the reviewer's comments have been highlighted in **red** in the revised manuscript and supporting information.

Reviewer #1

General Comments: *“In this work, the authors added formamide and guanidinium into an aqueous electrolyte to broaden the working temperature range and enhance thermopower. Au-coated carbon electrodes and a thermally insulating separator were employed to improve the electricity generation performance of the thermoelectric cell. There are some issues to be addressed before considering acceptance.”*

Response: We are grateful to the reviewer for the positive and encouraging comments. According to your constructive comments, we have carefully modified our manuscript. The corresponding revision has been listed as follows.

Comment #1: *“The authors need to explain the difference from the previously reported cryo-thermocell in terms of eutectic solvent strategy. (<https://doi.org/10.1002/adma.201901403>)”*

Response: Thanks for your comments.

#a. The reference (<https://doi.org/10.1002/adma.201901403>) is a work from Prof. X.T. Zhang, Chinese Academy of Sciences. In their work, X.T. Zhang *et al.*, for the first time, proposed the concept of the cryo ionic thermoelectric cell using this strategy and achieved a power output of 3.6 W m^{-2} under a temperature difference of 140 K ($T_c = -40^\circ\text{C}$).¹ However, in their work, the correlation between the eutectic behavior of the solvent and the thermoelectric performance of the cell was not clarified. In other words, detailed characterization is still lacking in the ionic thermoelectric materials field.

#b. In our work, we added this detailed analysis of the relative eutectic behavior of ionic thermoelectric materials, characterized by the high concentration of hydrated ions. Our work established a linear relationship between the freezing point of the solution and the lowest cold end temperature of the ionic thermoelectric materials through a series of performance tests, providing a more reference-worthy idea for the temperature range expansion of ionic thermoelectric systems.

#c. Our work was, for the first time, to combine the solvent eutectic strategy and the thermosensitive crystallization strategy to verify the feasibility of boosting the frigid temperature thermoelectric performance.

#d. We achieved a high transient power output of 17.5 W m^{-2} under a temperature difference of 105 K ($T_c = -35^\circ\text{C}$), which is nearly 3 times higher than the reference work. We further show the 2-hour energy output E_{2h} reached to 27 kJ m^{-2} , suggesting the practice value of ionic thermoelectric materials in the frigid temperature range.

Comment #2: *“The authors need to explain the difference from the previous paper that utilized thermally insulating separators from the standpoint of the thermal resistance effect. (<https://doi.org/10.1002/adma.201605652>)”*

Response: Thanks for your comments.

#a. The reference (<https://doi.org/10.1002/adma.201605652>) is a work from Prof. Ray H. Baughman *et al*, University of Texas at Dallas. In this work, Ray H. Baughman *et al*. originally introduced thermal insulating separator into the ionic thermoelectric electrolyte solution to increase thermal resistance.² However, in their work, the influence of electrolyte convection on thermal conductivity under temperature gradient has not been investigated. In other words, they only focused on the intrinsic thermal conductivity of electrolyte under isothermal conditions and ignored the equivalent thermal conductivity when convection was involved.

#b. In our work, our i-TE solution is more complicated, we have clarified the coupling effect between the thermosensitive crystallization and the thermal insulating separator. In our work, we found that the thermal insulating separator not only increases the overall thermal resistance of the cell, but also fixes the thermosensitive crystallization precipitation, prevents the precipitation from sinking due to gravity, and thus enables the cell to operate under various orientations of the temperature gradient. In other words,

our work, for the first time, show the thermal insulating separator could reduce the anisotropic of the i-TE solution cell. This is very different from the reference work.

#c. Furthermore, using the steady-state method,³ we measured the effective thermal conductivity of electrolyte and observed a significant reduction after adding a separator. We also examined how heat flux influenced the effective temperature difference and discovered that a separator could help the device reach a higher temperature difference under the same heat flux condition.

Comment #3: “Could you explain the difference from the previous paper using plasma-treated carbon electrodes regarding electrode wettability? (<https://doi.org/10.1002/adma.201303295>)”

Response: Thanks for your comments.

#a. The reference (<https://doi.org/10.1002/adma.201303295>) is a work from Prof. G. G. Wallace, University of Wollongong. In their work, G. G. Wallace *et al.* proposed to use the plasma treatment to carbon electrodes to improve their wettability for their ionic thermoelectric solution of $K_4Fe(CN)_6/K_3Fe(CN)_6$, in the experimental section without detail explain the reason.⁴

#b. In our work, we have conducted more insight characterization and analysis on the effect of wetting behavior of the electrolyte on the power output in our ionic solution. We have included voltage and output power density versus current density of the i-TE liquid cell (Figure 2b), Cyclic voltammograms (Figure 2d), Nyquist plots (Figure 2d), to show the wetting electrode is a critical factor to design the i-TE devices. The wetting electrode is important for accelerate the further diffusion of redox active ions into the internal microstructure of the electrode, and reduce the interface electrical resistance. Generally, we didn't intent to claim this plasmas treatment is our innovation, but

straightforward to show that it is need to pay more attention on the electrode wetting for higher power output.

Comment #4: “Could you provide the P_{\max} - T_c graphs (Figure 1d) of FA0 and FA50 after the cycle test shown in Figure 4f?”

Response: Thanks for your helpful suggestion. The P_{\max} - T_c graph of FA0 and FA50 after the cycle test have been added to the revised manuscript.

Revision: (Page 10 in the revised manuscript) “The P_{\max} - T_c graph of FA0 and FA50 after the one-week cycle test was also investigated. The P_{\max} of both exhibit varying degrees of attenuation (9.97 W m⁻² for FA50 and 10.2 W m⁻² for FA0), but the down-limit temperature ($T_{c,\min}$) remains constant (Figure S25).”

Revision: (in the revised supporting information)

Figure S25. Plot of maximum power P_{\max} of the FA50 ($h = 14$ mm) and FA0 ($h = 14$ mm) i-TE liquid cells versus cold-side temperature T_c (with fixed hot-side temperature $T_h = 70^\circ\text{C}$) after the one-week cycle test.

Comment #5: “Please compare the E_{2h} of FA0 ($T_c: 0^\circ\text{C}$) and FA50 ($T_c: -35^\circ\text{C}$).”

Response: Thanks for your comments. According to your suggestion, we have added the data of the energy output performance (E_{2h}) of FA0 to the revised manuscript.

Revision: (Page 10 in the revised manuscript) “The E_{2h} of FA0 ($T_c: 0^\circ\text{C}$) and FA50 ($T_c:-35^\circ\text{C}$) was also compared (Figure S21). The matching resistance with the highest E_{2h} of FA0 is smaller than that of FA50, indicating the lower internal resistance of FA0 ($T_c: 0^\circ\text{C}$). However, the smaller working temperature range of FA0 results in a lower working voltage and a lower E_{2h} , which compromise its output performance.”

Revision: (in the revised supporting information)

Figure S21. Long-term power generation of the FA0 i-TE liquid cell ($h = 10\text{ mm}$) at $T_c = 0^\circ\text{C}$ in i-TE generator working mode: a) Measured voltage and current curves during three working stages with a $20\ \Omega$ external resistance. b) Output power density measured over 2 h in stage (ii) using different external resistors. c) Corresponding generated energy density as a function of external resistors, calculated by integrating the output power over time (2 h) shown in (b).

Comment #6: “Why does the FA additive reduce the thermopower?”

Response: Thanks for the comments. In short, J. Zhou *et al.* has originally reported the addition of GdmCl enhanced thermopower of the solution of $\text{K}_4\text{Fe}(\text{CN})_6/\text{K}_3\text{Fe}(\text{CN})_6$, due to the chaotrope-chaotrope interaction between GdmCl and $[\text{Fe}(\text{CN})_6]^{4-}$. In our work, the addition of FA could weaken the chaotrope-chaotrope interaction and hence reduce the thermopower. More detail discussions are given as follows.

#a. We have added the UV-Vis absorption spectra of the FA-free $\text{K}_4\text{Fe}(\text{CN})_6$ solution with and without GdmCl (Figure S13a), and the $\text{K}_4\text{Fe}(\text{CN})_6$ -GdmCl solution with different FA contents (Figure S13b) to characterized the interactions between GdmCl

and $[\text{Fe}(\text{CN})_6]^{4-}$. First, the absorption peak for $[\text{Fe}(\text{CN})_6]^{4-}$ shifts significantly from 178 nm to 209 nm after the addition of GdmCl, which reveals the chaotrope-chaotrope interaction between GdmCl and $[\text{Fe}(\text{CN})_6]^{4-}$. This is consistent with J. Zhou's work.³ However, in Figure S13b, the absorption peak for $[\text{Fe}(\text{CN})_6]^{4-}$ of the i-TE solution of $\text{K}_4\text{Fe}(\text{CN})_6$ -GdmCl (FA/Water) reversely move from 209 nm to 184 nm. As a result, it is believed that the chaotrope-chaotrope interactions are gradually weakened by the addition of the FA.

#b. Furthermore, we also added the comparison of thermopower of two serials i-TE solution, i.e. 0.4M $\text{FeCN}^{4-/3-}$ and 0.4M $\text{FeCN}^{4-/3-}$ -3M GdmCl with different FA (Figure S14). It is shown that, in the GdmCl-free case, the thermopower is almost unchanged with FA. In contract, in the 0.4M $\text{FeCN}^{4-/3-}$ -3M GdmCl, the thermopower decreased with increased FA. This new result suggested that the FA might impact the hydrate structure GdmCl rather the $[\text{Fe}(\text{CN})_6]^{4-}$, and finally impact the interaction between GdmCl and $[\text{Fe}(\text{CN})_6]^{4-}$. We have added the relevant instructions to the revised manuscript.

Revision: (Page 6 in the revised manuscript) “FA weakens the chaotropic-chaotropic interaction between GdmCl and $[\text{Fe}(\text{CN})_6]^{4-}$, which lowers the reaction entropy of the redox couples and thus the thermopower. The UV-Vis spectra of FA-free $\text{K}_4\text{Fe}(\text{CN})_6$ with and without GdmCl (Figure S13a), and $\text{K}_4\text{Fe}(\text{CN})_6$ -GdmCl with varying FA (Figure S13b) were obtained to characterize the GdmCl- $[\text{Fe}(\text{CN})_6]^{4-}$ interaction. First, the $[\text{Fe}(\text{CN})_6]^{4-}$ peak shifts from 178 nm to 209 nm upon GdmCl addition, indicating the chaotropic-chaotropic interaction between GdmCl and $[\text{Fe}(\text{CN})_6]^{4-}$. However, in Figure S13b, the $[\text{Fe}(\text{CN})_6]^{4-}$ peak of the i-TE solution of $\text{K}_4\text{Fe}(\text{CN})_6$ -GdmCl (FA/Water) reverses from 209 nm to 184 nm, indicating that the chaotropic-chaotropic interactions are weakened by FA. Moreover, we compared the thermopower of two i-TE solution series, namely 0.4 M $\text{FeCN}^{4-/3-}$ and 0.4M $\text{FeCN}^{4-/3-}$ -3M GdmCl with different FA (Figure S14). It shows that, without GdmCl, the thermopower is nearly constant with FA. In contrast, in 0.4M $\text{FeCN}^{4-/3-}$ -3M GdmCl, the thermopower decreases with

increasing FA. This suggests that FA affects the hydrate structure of GdmCl rather than $[\text{FeCN}_6]^{4-}$, and consequently the GdmCl- $[\text{FeCN}_6]^{4-}$ interaction.”

Revision: (in the revised supporting information)

Figure S13. UV-Vis spectral shifts and the corresponding absorption peaks for $[\text{FeCN}_6]^{4-}$. a) UV-Vis spectra for FA-free $\text{K}_4\text{Fe}(\text{CN})_6$ solution with and without GdmCl, and b) $\text{K}_4\text{Fe}(\text{CN})_6$ -GdmCl solution with different FA contents.

Figure S14. Thermopower of 0.4M $\text{FeCN}^{4-/3-}$ and 0.4M $\text{FeCN}^{4-/3-}$ -3M GdmCl with different FA contents.

Comment #7: “What is the voltage sweep rate when measuring performance? (e.g., Fig. 1d) Does it affect the performance?”

Response: Thanks for the comments. We would like to address your comments separately.

a. *What is the voltage sweep rate when measuring performance? (e.g., Fig. 1d)*

The current–voltage characterization of the device was performed with a Keithley 2400 instrument. The Signal acquisition rate is 0.1 s per point. There are approximately 50 points between 0 V to open-circuit voltage. Thus the voltage sweep rate is about 60 mV s⁻¹.

b. *Does it affect the performance?*

For the test of instantaneous output power, a faster voltage sweep rate is usually adopted to examine the transient behavior of active ions at the electrode. The curve obtained in this case is called a voltammetric curve, and the current includes both Faradaic and non-Faradaic currents. If the scan rate is too slow, the electrode surface is considered to be in a steady state, and the non-Faradaic current disappears. Here, by changing the number of acquisition points (10, 30, 50, 70, and 90 points) between 0 V to open-circuit voltage, the voltage-current curves under different voltage sweep rates (300, 100, 60, 43, 33 mV s⁻¹) were measured (Figure R1). It can be found that P_{\max} fluctuates slightly under different voltage sweep rates. The value of P_{\max} at the sweep rate selected in this paper is at the average level, which is of reference significance. We have added the relevant instructions to the revised manuscript.

Figure R1. a) Voltage and b) output power density versus current density for the as-fabricated i-TE liquid cell of FA50 ($h = 10\text{mm}$, $T_h = 70^\circ\text{C}$, $T_c = -35^\circ\text{C}$) under different

voltage sweep rates (300, 100, 60, 43, 33 mV s⁻¹). c) The graph of output power density versus voltage sweep rate (300, 100, 60, 43, 33 mV s⁻¹).

Revision: (Page 13 in the revised manuscript) “The voltage was measured with a Keithley-2000 instrument and the current–voltage characterization of the device was performed with a Keithley 2400 instrument. There are approximately 100 points between 0 V to open-circuit voltage. The voltage sweep rate is 0.1 s per point.”

Comment #8: *“Please provide the voltage and output power density versus current density for Figure 3d in the supporting information.”*

Response: Thanks for your helpful suggestion. We have added the detailed data for Figure 3d in the supporting information.

Revision: (in the revised supporting information)

Figure S17 Voltage and output power density versus current density for the as-fabricated Hydrophilic Au@CP|H₂O/FA-FeCN^{4-/3-}-GdmCl (FA50)|Hydrophilic Au@CP i-TE liquid cell under different T_c with a fixed $T_h = 70$ °C. a) and b) $h = 14$ mm, $T_c = -30, -35,$ and -40 °C. c) and d) $h = 12$ mm, $T_c = -30, -35,$ and -40 °C. e) and f) $h = 10$ mm, $T_c = -30, -35,$ and -40 °C. g) and h) $h = 8$ mm, $T_c = -30, -35,$ and -40 °C.

Comment #9: “It is encouraged to add Fig. 3a and b for FA50 results.”

Response: Thanks for your helpful suggestion. The temperature differences versus heat flux in three electrolyte configurations for FA50 have added in these figures.

Revision:

(Page 8 in the revised manuscript) “Figure 3a plots the effective ΔT across the i-TE liquid cells with and without the TIS as functions of heat flux. The tested cells were $\text{H}_2\text{O}/\text{FA}-0.4\text{M FeCN}^{4-/3-}$, $\text{H}_2\text{O}/\text{FA}-0.4\text{M FeCN}^{4-/3-}-3\text{M GdmCl}$ (FA50), and FA50 with TIS.”

(Page 8 in the revised manuscript) “Furthermore, the TIS desensitized the i-TE liquid cell to the temperature-gradient direction (Figure 3b), which affects the power generation of the cell.”

(Page 21 in the revised manuscript)

Figure 3. a) Temperature differences between the cold and hot electrodes as functions of heat flux in three electrolyte configurations (FA50 with neither GdmCl nor a thermal insulation separator (TIS), FA50 with GdmCl but without TIS, FA50 with both GdmCl and TIS). b) P_{max} of the i-TE liquid cell in three electrode orientations with or without the TIS.

Comment #10: “Does a FA added electrolyte have a constant thermopower over a range of temperatures ($T_{c,\text{min}} \sim T_h$)?”

Response: Thanks for the comments. We have added the original data of thermopower measurements in the supporting materials (Figure S12). According to the linear fitting results, all the electrolytes (FA0~FA60) maintain stable thermopower within a large temperature difference range. However, when the temperature difference exceeds the

limit, the thermopower drops sharply, indicating the failure of the electrolyte at that temperature.

Revision: (in the revised supporting information)

Figure S12. a) Dependence of the voltage difference ($V_c - V_h$) on the temperature difference ($T_h - T_c$, $T_h = 70$ °C) for the as-fabricated i-TE liquid cell of H_2O/x FA-0.4M $FeCN^{4-/3-}$ -3M GdmCl ($h = 14$ mm) ($x = 0$ vol%, 10 vol%, 20 vol%, 30 vol%, 40 vol%, 50 vol%, 60 vol%); b) Thermopower versus the content of FA.

Reviewer #2

Comment #1: *“In the introduction part, the eutectic solvent strategy should be further introduced to highlight their advantages in the possible application of cryo-temperature region.”*

Response: Thank you for your comments. We have revised the introduction part of our paper to include more details about the eutectic solvent strategy and its advantages in the possible application of cryo-temperature region.

Revision: (Page 3 in the revised manuscript) *“Benefiting from the natural of eutectic solvents that form a liquid mixture at a lower melting point than that of any individual component, electrolytes consisting of a eutectic mixture solvent such as Formamide (FA)/H₂O,⁵ Ethylene glycol (EG)/H₂O,^{6,7} and Dimethyl sulfoxide (DMSO)/H₂O,⁸ were promising candidates for cryogenic processes in energy-related systems. Given the ability of eutectic solvents to offer superior ionic transport conditions for electrolytes in subzero temperatures, they present a viable approach to lower the minimum operating temperature of ionic thermoelectric cells into the cryogenic range.”*

Comment #2: *“As demonstrated, the wettability of CP electrode by electrolyte plays a significant role to enhance the thermoelectrochemical performance such as power density. However, Figure S13 only provides the contact angle of hydrophobic CP (hp-CP) and hydrophilic CP (hl-CP). In my view, the contact angle of used hl-Au@CP also should be provided.”*

Response: Thanks for the comments. I agree that the contact angle of used hl-Au@CP is important to show the effect of wettability on the thermoelectrochemical performance. Therefore, according to your suggestion, we added the contact angle test of the hl-Au@CP electrode.

Revision: (in the revised supporting information)

Figure S15. Contact angle of the ionic solution (FA50) on a) hydrophobic CP, b) hydrophilic CP, and c) hydrophilic Au@CP electrode is 114.6°, 0° and 0°, which means that the electrolyte has better wettability to the hydrophilic Au@CP electrode.

Comment #3: “To my knowledge, the thermal conductivity of electrolyte and electrode affect the performances of *i*-TEs. Please supplement some related results.”

Response: Thanks for the comments. The thermal conductivity of the electrode and the electrolyte determines the effective temperature difference that the cell can generate under a certain heat flux. The lower the thermal conductivity, the larger the effective temperature difference, and the higher the thermoelectric voltage. Generally speaking, for thermogalvanic cells, the temperature at the interface between the electrode and the electrolyte is the temperature at which the reaction occurs. To increase the effective temperature difference, we hope that the electrode has a higher thermal conductivity, while the electrolyte has a lower thermal conductivity. In this paper, carbon paper is used as the electrode, and graphite plate is used as the current collector, both of which have excellent thermal conductivity. The hot-disk method is used to measure their thermal conductivity. The test results are presented in Table S1.

For liquid electrolytes, due to their fluidity, when temperature is applied, the thermal conductivity is contributed by both heat conduction and convection. This makes their effective thermal conductivity significantly higher than their intrinsic thermal conductivity under isothermal conditions. Therefore, the thermal conductivity measurement method for liquid electrolytes should be able to directly measure the effective thermal conductivity, rather than the intrinsic thermal conductivity. Here, we use a steady-state method to measure the effective thermal conductivity.³ As shown in

Figure R2, the test cell consisted of two identical graphite plates (a thickness (d_1) of 3 mm and a thermal conductivity (κ_1) of $0.16 \text{ W m}^{-1} \text{ K}^{-1}$) in parallel at the top and bottom as heat transfer walls, with a distance (d_2) of 1.5 cm. An electrical heating plate contacted the graphite wall at the bottom. The electrolyte with a thermal conductivity of κ_{eff} were filled into the middle cell. The cross-sectional area of the middle cell was 2.6 cm^2 . In order to ensure that the input thermal flow (Q_{input}) is equal to the output thermal flow (Q_{output}), the side walls of the planar cell were sealed by thin polyethylene terephthalate (PET) sheets with a thickness of 0.5 mm and then covered by polystyrene foam with thickness of 2 cm to prevent heat dissipation into the surroundings. Therefore, the heat flow across the two plate walls is equal to the heat flow across the middle electrolyte, which is defined as: $\kappa_1 A \left(\frac{\partial T}{\partial d} \right)_1 = \kappa_{\text{eff}} A \left(\frac{\partial T}{\partial d} \right)_2$. According to this Eq., if the steady-state temperature gradient ($\partial T/\partial d$) is achieved across the graphite walls and electrolyte, then we can calculate the thermal conductivity of the electrolyte. A thermocouple (Omega, TT-T-36-SLE) was placed at each of the four positions shown in the figure to monitor the temperature. The test results are shown in Table S1.

Figure R2. The testing device of the steady-state method to measure the effective thermal conductivity of liquid electrolytes.

Revision: (Page 8 in the revised manuscript) “Second, the TIS effectively suppresses heat convection and increases the thermal resistance of the cell (The thermal conductivity of the electrolyte and electrode are shown in Table S1).”

Revision: (Page 13 in the revised manuscript) “The thermal conductivity of the electrode was measured with hot-disk method, and the effective thermal conductivity of the electrolyte was measured with the steady-state method³.”

Revision: (in the revised supporting information)

	Thermal conductivity (W m ⁻¹ K ⁻¹)
Graphite current collector	125
hp-CP	1.8
hl-CP	1.8
hl-Au@CP	1.7
H ₂ O/FA-0.4M FeCN ^{4-/3-}	1.6
FA50	0.4
FA50 with TIS	0.3

Table S1. Thermal conductivity of electrolyte and electrode.

Comment #4: “As shown in Figure 4f, why does FA50 i-TE exhibit a voltage decay with the one-week continuous cycling? By the way, in the caption of Figure 4f, the first five days should be seven days? Same question also involved in the related Figure S19.”

Response: Thanks for the comments. We would like to address your comments separately.

a. As shown in Figure 4f, why does FA50 i-TE exhibit a voltage decay with the one-week continuous cycling?

Thanks for your comments. The degradation of cyclic performance is caused by many complex factors. Here, we speculate that there may be two main reasons for the degradation of cyclic performance: first, the thermosensitive crystallization dissolution kinetics gradually slows down, and the active substances that can participate in the electrode reaction decrease. The thermosensitive crystallization is exposed to a high temperature environment for a long time, continuously dissolving and forming new active ions, and the active ions diffusing from the cold end form new crystallization. This process is repeated, resulting in the continuous decay of the dissolution

performance of the crystallization. In this way, the insufficient supply of active ions required for the electrode reaction will lead to the degradation of performance. Second, the formation and dissolution process of thermosensitive crystallization has a long-term impact on the electrode. With the progress of the charge-discharge process and the continuous dissolution and formation of the crystallization, the electrode surface is gradually covered with a passivation film (The optical microscope image of the electrode before and after cycling test was shown in Figure S23, and the specific composition of the film will be discussed in Comment #5). This film not only increases the ohmic resistance of the electrode, but also hinders the further diffusion and reaction of active particles on the electrode, thereby leading to the degradation of performance. We have added relevant explanations in the revised manuscript.

Revision: (Page 10 in the revised manuscript) “The 2 h energy density (E_{2h}) decreased from 27 kJ m^{-2} on day 1 to 15 kJ m^{-2} on day 7, for which the following reasons may account: first, the thermosensitive crystallization dissolution kinetics were presumably slowed by long cycle service at $T_h = 70^\circ\text{C}$, resulting in the insufficient supply of active ions required for the electrode reaction. Second, the formation and dissolution process of thermosensitive crystallization has a long-term impact on the electrode, such as the electrode surface is gradually covered with a passivation film (Figure S23) which hinders the further diffusion and reaction of active particles on the electrode.”

Figure S23. Optical microscope image of the electrodes a) before and b) after cycling test at $T_h = 70^\circ\text{C}$. With the one-week continuous cycling test, the electrode surface is gradually covered with a passivation film.

b. *By the way, in the caption of Figure 4f, the first five days should be seven days?*

Thanks for the comments. We have corrected the description errors in the relevant parts of the paper.

c. *Same question also involved in the related Figure S19.*

Thanks for the comments. We have corrected the description errors in the relevant parts of the paper.

Revision: (Page 10 in the revised manuscript) “The measured voltage and current curves of the build-up voltage and power-output stages during the first seven days and the output power density during the second stage on the first, third, fifth, and seventh days are shown in Figure S22.”

Revision: (Page 22 in the revised manuscript) “f) Measured voltage and current curves during one week of continuous cycling ($T_h = 50^\circ\text{C}$).”

Revision: (in the revised supporting information) “Figure S22. a) The measured voltage and current curves of build-up voltage and power output stages for the first seven days.”

Comment #5: “The morphology of electrodes after cycling test should be provided.”

Response: Thanks for the comments. As comment 4 stated, after the cyclic test, the hot-end electrode surface was coated with a non-uniform passivation layer, which reduced the cyclic performance. The Optical microscope image of the electrodes after cycling test at $T_h = 70^\circ\text{C}$ was shown as **Figure S23b** in Comment #4. We also characterized the specific composition of the passivation layer covering the electrode with XRD and XPS measurement (**Figure S24**). The results show that the passivation layer is essentially the

thermosensitive crystal formed by the combination of GdmCl and $\text{Fe}(\text{CN})_6^{4-}$. During the cyclic testing process, the grain size gradually decreases and easily adheres to the electrode surface to form the passivation layer. We have added this content to the revised manuscript.

Revision: (Page 10 in the revised manuscript) “The specific composition of the passivation layer covering the electrode was also characterized with XRD (Figure S24a) and XPS (Figure S24b) measurement. It can be observed that the passivation layer is essentially the thermosensitive crystal formed by the combination of GdmCl and $\text{Fe}(\text{CN})_6^{4-}$. During the cyclic testing process, the grain size gradually decreased and easily adhered to the electrode surface to form the passivation layer, resulting in the performance degradation.”

Revision: (in the revised supporting information)

Figure S24. a) The XRD spectra, and b) N 1s XPS spectra comparison of pure $\text{K}_3\text{Fe}(\text{CN})_6$, $\text{K}_4\text{Fe}(\text{CN})_6$, GdmCl powders, and the formed passivation layer, showing their characteristic peaks.

Comment #6: “Some format type in the references should be corrected, such as Ref. 3, 21, 36.”

Response: Thanks for the comments. We have corrected the format type of Ref. 3, 21, and 36.

Revision: (Page 14 in the revised manuscript) “3. Eastman, E. D. Theory of the Soret Effect. *J. Am. Chem. Soc.* **50**, 283–291 (1928).”

Revision: (Page 15 in the revised manuscript) “21. Iwami, R., Yamada, T. & Kimizuka, N. Increased Seebeck Coefficient of $[\text{Fe}(\text{CN})_6]^{4-/3-}$ Thermocell Based on the Selective Electrostatic Interactions with Cationic Micelles. *Chem. Lett.* (2020).”

Revision: (Page 17 in the revised manuscript) “41. Su-Moon Park and Jung-Suk Yoo. Peer Reviewed: Electrochemical Impedance Spectroscopy for Better Electrochemical Measurements. *Anal. Chem.* **75**, 455 A-461 A (2003).”

Reference

1. Li, G. *et al.* High-Efficiency Cryo-Thermocells Assembled with Anisotropic Holey Graphene Aerogel Electrodes and a Eutectic Redox Electrolyte. *Advanced Materials* **31**, 1901403 (2019).
2. Zhang, L. *et al.* High Power Density Electrochemical Thermocells for Inexpensively Harvesting Low-Grade Thermal Energy. *Adv. Mater.* **29**, 1605652 (2017).
3. Duan, J. *et al.* Aqueous thermogalvanic cells with a high Seebeck coefficient for low-grade heat harvest. *Nat Commun* **9**, 5146 (2018).
4. Romano, M. S. *et al.* Carbon Nanotube - Reduced Graphene Oxide Composites for Thermal Energy Harvesting Applications. *Adv. Mater.* **25**, 6602–6606 (2013).
5. Gao, Y., Qin, Z., Guan, L., Wang, X. & Chen, G. Z. Organoaqueous calcium chloride electrolytes for capacitive charge storage in carbon nanotubes at sub-zero-temperatures. *Chem. Commun.* **51**, 10819–10822 (2015).
6. Roberts, A. J., Namor, A. F. D. de & Slade, R. C. T. Low temperature water based electrolytes for MnO₂/carbon supercapacitors. *Phys. Chem. Chem. Phys.* **15**, 3518–3526 (2013).
7. Chang, N. *et al.* An aqueous hybrid electrolyte for low-temperature zinc-based energy storage devices. *Energy Environ. Sci.* **13**, 3527–3535 (2020).
8. Nian, Q. *et al.* Aqueous Batteries Operated at –50 °C. *Angewandte Chemie International Edition* **58**, 16994–16999 (2019).

REVIEWER COMMENTS

Reviewer #1 (Remarks to the Author):

I am satisfied with the work the authors did to address the questions. However, there are still several points to clarify.

(1) The authors are encouraged to introduce the previously reported papers for cryo-thermocell and thermally insulating separators in the main text, which will help readers easily grasp the related research.

(2) After one week of cycling, corresponding to the power generation of 14 hours (Fig. 4f), FA50 showed a more severe degradation than FA0 as shown in Fig. S25. As a result, despite the extended working temperature range of FA50, it made a power rather smaller than FA0 did. The authors need to give more discussion on the result.

(3) During each one-day cycle, self-reactivation time is 4 hours and rest time is 18 hours. How do they affect the performance?

(4) In the discussion for Fig. 3(d), how does the increased inter-electrode heat transport shift the inter-electrode temperature drop toward the hot electrode? Considering a constant open-circuit voltage, the temperature difference between electrodes might be unchanged.

(5) In figure S17, the cell's internal resistance decreased as h increased from 14 to 10 mm and increased again at $h = 8$ mm. It is suggested to provide a more detailed explanation of the results.

(6) What are the thickness and density of TIS?

(7) Introduction

1. Li et al. introduced glutaraldehyde to gelatin, forming strong covalent bonds that increased the $T_{h,max}$ from 9°C to 23°C while maintaining $T_{c,min}$ at 21°C.

2. The maximum temperature range of an i-TE liquid cell is limited by the freezing or boiling point of the solvent and the reported $T_{c,min}$ values of i-TE liquid cells always exceed 0°C.

These need to be corrected.

(8) Please check the caption of Figure S5.

(9) 4.3 Fabrication of the i-TE liquid cell: A cotton-fiber laminating thermal separator was fabricated by compressing commercial cotton balls with approximate diameters of 20 μm (Figure S7).

Please check the figure number.

Reviewer #2 (Remarks to the Author):

The authors have revised their manuscript according to the comments. I think it should be accepted for publication.

Response to reviewers

Nature Communication

Manuscript ID: NCOMMS-23-44213

Title: High-Performance Cryo-Temperature Ionic Thermoelectric Liquid Cell

Developed through a Eutectic Solvent Strategy

Dear Reviewers:

We would like to thank you for your kindness in offering us the opportunity to improve our manuscript as per the valuable comments, which provided important insights to improve the quality of this work. Herein, we provide a detailed point-by-point response to the comments. In the following passages, comments are in *italics*, followed by our response in black. Revisions in response to the reviewer's comments have been highlighted in red in the revised manuscript and supporting information.

Reviewer #1

General Comments: *“I am satisfied with the work the authors did to address the questions. However, there are still several points to clarify.”*

Response: We are grateful to the reviewer for the positive and encouraging comments. According to your constructive comments, we have carefully modified our manuscript. The corresponding revision has been listed as follows.

Comment #1: *“The authors are encouraged to introduce the previously reported papers for cryo-thermocell and thermally insulating separators in the main text, which will help readers easily grasp the related research.”*

Response: Thanks for your comments. We have introduced the previously reported works in the main text.

Revision: (Page 3 in the revised manuscript) *“For instance, Li *et al.* assemble a cryo-thermocell with a eutectic redox electrolyte of formamide and water and achieved a P_{\max} of 3.6 W m^{-2} at $\Delta T = 106^\circ\text{C}$.”*

Revision: (Page 8 in the revised manuscript) *“For instance, by using a hydrophilic cellulose sponge, Baughman *et al.* improved the ΔT of the cell from 63 to 95°C .”*

Comment #2: *“After one week of cycling, corresponding to the power generation of 14 hours (Fig. 4f), FA50 showed a more severe degradation than FA0 as shown in Fig. S25. As a result, despite the extended working temperature range of FA50, it made a power rather smaller than FA0 did. The authors need to give more discussion on the result.”*

Response: Thanks for your comments.

#a. We conducted a detailed comparison of the long-term power generation of the FA50 ($T_h = 70^\circ\text{C}$, $T_c = -35^\circ\text{C}$) (Figure S26) and FA0 ($T_h = 70^\circ\text{C}$, $T_c = 0^\circ\text{C}$) (Figure S26) i-TE liquid cells during one week of cyclic service.

#b. The decay of two-hour energy output E_{2h} of FA50 and FA0 during one week of cyclic service were also compared (Figure S26c). Despite exhibiting more severe degradation than FA0, FA50 still had a higher total power capacity of 14 hours (152.4 kJ m^{-2} for FA50 compared to 142.2 kJ m^{-2} for FA0).

#c. The instantaneous output power (P_{\max}) of the cell was affected by the long-term cyclic service, and the P_{\max} of FA50 ($T_h = 70^\circ\text{C}$, $T_c = -35^\circ\text{C}$) and FA0 ($T_h = 70^\circ\text{C}$, $T_c = 0^\circ\text{C}$) after seven days of cycling were given by Figure S27. The P_{\max} of both exhibit varying degrees of attenuation (9.97 W m^{-2} for FA50 and 10.2 W m^{-2} for FA0). The lower P_{\max} of FA50 was attributed to its poorer cycling performance, which was influenced by factors such as larger working temperature difference, lower cold-end temperature, lower conductivity and higher viscosity of FA50 solution.

We have discussed these results in more detail in the revised manuscript.

Revision: (Page 11 in the revised manuscript) “The cyclic working–resting performance of FA0 ($T_h = 70^\circ\text{C}$, $T_c = 0^\circ\text{C}$) was also conducted (Figure S26). The decay of two-hour energy output E_{2h} of FA50 and FA0 during one week of cyclic service were compared (Figure S26c). Despite exhibiting more severe degradation than FA0, FA50 still had a higher total power capacity of 14 hours (152.4 kJ m^{-2} for FA50 compared to 142.2 kJ m^{-2} for FA0). The instantaneous output power (P_{\max}) of the cell was affected by the long-term cyclic service, and the P_{\max} - T_c graph of FA0 and FA50 after the one-week cycle test was given by Figure S27. The P_{\max} of both exhibit varying degrees of attenuation (9.97 W m^{-2} for FA50 and 10.2 W m^{-2} for FA0). The lower P_{\max} of FA50 was attributed to its poorer cycling performance, which was influenced by factors such as larger working temperature difference, lower cold-end temperature, lower conductivity and higher viscosity of FA50 solution.”

Revision: (in the revised supporting information)

Figure S26. Long-term power generation of the *hl*-Au@CP|FA0-TIS|*hl*-Au@CP i-TE liquid cell ($h = 10$ mm) during one week of continuous cycling ($T_h = 70^\circ\text{C}$, $T_c = 0^\circ\text{C}$). a) The measured voltage and current curves of build-up voltage and power output stages for the first seven days and b) the output power density of the second stage on the first, third, fifth, and seventh day. c) Decay of two-hour energy output E_{2h} of the FA0 ($T_h = 70^\circ\text{C}$, $T_c = 0^\circ\text{C}$) and FA50 ($T_h = 70^\circ\text{C}$, $T_c = -35^\circ\text{C}$) i-TE liquid cells during one week of cyclic service.

Comment #3: “During each one-day cycle, self-reactivation time is 4 hours and rest time is 18 hours. How do they affect the performance?”

Response: Thanks for your comments. To address this issue, we compared the cycling performance of the FA50 i-TE liquid cell ($T_h = 70^\circ\text{C}$, $T_c = -35^\circ\text{C}$) under different recovery modes: self-reactivation with short circuiting at $\Delta T = 0$ K (4 h) and rest (18 h); self-reactivation (4 h) without rest; rest (18 h) without self-reactivation. As shown in Figure R1, the cycling performance measured under the three recovery modes was almost identical, which proved that the choice of recovery mode (self-reactivation or rest) was not important for the cycling performance under the condition of sufficient recovery time. We have added relevant description in the revised manuscript.

Figure R1. Decay of two-hour energy output E_{2h} of the FA50 ($T_h = 70^\circ\text{C}$, $T_c = -35^\circ\text{C}$) i-TE liquid cells during cyclic service with different recovery modes.

Revision: (Page 10 in the revised manuscript) “The time of reactivation was set to 4 hours to ensure that the open-circuit voltage returned to zero”

Comment #4: “In the discussion for Fig. 3(d), how does the increased inter-electrode heat transport shift the inter-electrode temperature drop toward the hot electrode? Considering a constant open-circuit voltage, the temperature difference between electrodes might be unchanged.”

Response: Thanks for your helpful comments. In the original manuscript, we attempted to explain the abnormal performance of the cell at $h = 8$ mm by the non-uniformity of the temperature gradient. The expression in the manuscript may not be clear enough, and it is re-explained here: When the temperature difference between the cold and hot electrodes is constant, the reduction of the electrode spacing may cause the temperature gradient in the height direction to change. Due to the non-uniformity of the temperature gradient in the height direction, we speculate that when the electrode spacing decreases, the temperature gradient near the hot end electrode changes more sharply, which will cause the average temperature inside the battery to decrease, thereby reducing the average ionic conductivity and affecting the short-circuit current. We acknowledge that the explanation in the original manuscript was too rough, so we conducted a more

detailed study on the effect of electrode separation gap (h), which will be discussed in detail in comment #5.

Comment #5: *“In figure S17, the cell’s internal resistance decreased as h increased from 14 to 10 mm and increased again at $h = 8$ mm. It is suggested to provide a more detailed explanation of the results.”*

Response: Thanks for your comments. In order to examine the impact of electrode separation gap in detail, we simulated the state of the solution using COMSOL Multiphysics 6.0 with a comprehensive model that coupled the mass and heat transfer and fluid flow when the cell with varying electrode separation gaps functioned under a 105°C temperature gradient ($T_h = 70^\circ\text{C}$, $T_c = -35^\circ\text{C}$).

#a. The natural convection caused by the temperature difference has an important influence on the performance. As shown in Figure 3d, as the electrode gap decreases, the speed of natural convection is significantly weakened, which will slow down the circulation speed of the redox pair between the cold and hot electrodes. The reactants of the thermogalvanic reaction cannot be replenished in time, and the reaction products cannot be transferred in time, thus increasing the mass transport overpotential and leading to the performance degradation.

#b. The temperature distribution between the electrodes at different electrode separation gap was also simulated. As shown in Figure S19, there is a non-uniform temperature gradient across the electrodes, and a temperature gradient also exists horizontally, which was caused by the convection and the non-uniformity of the local thermal resistance. The temperature drop near the hot end electrode gradually becomes more severe as the h decreases, and the average temperature of the cell as a whole is significantly lower.

#c. The influence of convection on the i-TE liquid cell is a complex proposition. On the one hand, we hope to reduce the equivalent thermal conductivity of the cell by

suppressing the strong natural convection of the solution under a large temperature difference. On the other hand, we also hope that the natural convection is intense enough to accelerate the mass transfer and circulation of the thermogalvanic reaction. And there are various factors affecting convection. When the temperature difference is determined, the smaller the electrode gap, the slower the convection, but the ohmic overpotential is decreasing; the introduction of the TIS will also slow down the convection speed, but it will also bring a higher overall temperature to the solution. In a word, the optimization of the structure always faces the trade-off and restriction of various factors, and we are just finding a balance point. We have updated the relevant explanations in the manuscript.

Revision: (Page 8 in the revised manuscript)

The shorter of the electrode separation gap would have less internal resistance, hence resulting in the increase I_{sc} . In order to clarify the abnormal decreased I_{sc} as the h further decreased to 8 mm, we have conducted simulation with COMSOL Multiphysics 6.0 to investigate the effect of electrode separation gap on the thermogalvanic convection. In the i-TE solution of $K_3Fe(CN)_6/K_4Fe(CN)_6$, the ionic convection is due to the accumulated $Fe(CN)_6^{3-}$ from oxidation of $Fe(CN)_6^{4-}$ at hot side, resulting into back flux of $Fe(CN)_6^{3-}$ from hot side to cold, and vice versa for cold side. The convection of redox couples also afford the i-TE cells work continuous, and also have higher energy output than that with only thermodiffusion ions. Figure 3d compares the simulated convection of the i-TE cell of $H_2O/FA-0.4M FeCN^{4-/3-}-3M GdmCl$ (FA50) with different electrode gap, and a fixed temperature difference of $105^\circ C$ ($T_h = 70^\circ C$, $T_c = -35^\circ C$). It clearly shows that the electrode gap has a significant effect on the convection speed, showing that the redox pair circulation slow down as the electrode separation gap decreases. This reduced convection would result in a decreased I_{sc} . In other words, the decreased h has two opposite effects on the I_{sc} . The balance between the reduced internal resistance and suppressed convection results in the optimized h . Additionally, we also noted that the convection of redox couple also resulted in the non-uniform temperature gradient both vertically and horizontally (Figure S19), which add the

complex to understand the contribution of the microscopic ionic behavior to macroscopic thermovoltage.

Revision: (Page 22 in the revised manuscript)

Figure 3. Structural optimization of the H₂O/FA-FeCN^{4-/3-}-GdmCl i-TE liquid cell: a) Temperature differences between the cold and hot electrodes as functions of heat flux in three electrolyte configurations (FA50 with neither GdmCl nor a thermal insulation separator (TIS), FA50 with GdmCl but without TIS, FA50 with both GdmCl and TIS). b) P_{\max} of the i-TE liquid cell in three electrode orientations with or without the TIS. c) Open circuit voltage V_{oc} , short-circuit current I_{sc} , and P_{\max} in the H₂O/FA-FeCN^{4-/3-}-GdmCl i-TE liquid cell with different electrode separations gaps. d) Simulated convection profiles of the FA50-TIS i-TE liquid cell with varying electrode separations

gaps at $T_h = 70^\circ\text{C}$ and $T_c = -35^\circ\text{C}$. e) I_{sc} , P_{max} , and effective resistance R_{eff} of the FA50 cell ($h = 10$ mm) versus T_c . f) Comparison of P_{max} and $T_{c,min}$ between the proposed and reported i-TE liquid cells.

Revision: (in the revised supporting information)

Figure S19. The simulated temperature distribution between the electrodes of the FA50-TIS i-TE liquid cell at different electrode separation gaps (h). a) $h = 14$ mm, b) $h = 12$ mm, c) $h = 10$ mm, d) $h = 8$ mm. COMSOL Multiphysics 6.0 with a comprehensive model coupling mass and heat transfer and fluid flow was used to simulate the convection and temperature distributions of the solution in the FA50-TIS cell at $T_h = 70^\circ\text{C}$ and $T_c = -35^\circ\text{C}$. The cell model was vertically oriented without the use of forced convection, and the temperature difference was built from the bottom (hot) to the top (cold). This free convection problem was modeled by introducing the Boussinesq buoyancy term into the Brinkman momentum equation, and then coupling the resulting fluid velocity with the “Heat Transfer in Porous Media” interface. In addition, the parameters used in the simulation are summarized in Table S2.

Revision: (in the revised supporting information)

Table S2. Calculation parameters for COMSOL simulation.

Parameter	Value
Cell radius	1cm
Hot bottom temperature	343K
Cold top temperature	238K
Solution density	1096.7 kg m ⁻³
Kinematic viscosity	0.6×10 ⁻⁶ m ² s ⁻¹
Solution specific heat capacity	4187 J kg ⁻¹ K ⁻¹
Solution thermal conductivity (κ)	0.55 W m ⁻¹ K ⁻¹
Separator porosity	0.3
Separator permeability	2.35m ²
Sediment porosity	0.16
Sediment permeability	1.29m ²

Comment #6: “What are the thickness and density of TIS?”

Response: Thanks for the comments.

For cell with different electrode separation gaps (h), the thickness of TIS is different. The thickness of TIS is set to half of h , for example, a 14 mm thick cell uses a 7 mm thick TIS, and a 10 mm thick cell uses a 5 mm thick TIS. The density of TIS is 0.15 g cm⁻³.

We have added this information to the corresponding position in the manuscript.

Revision: (in the revised supporting information)

Figure S16. a) Entity profile and b) schematic drawing of the thermal insulation separator (TIS) made by cotton fiber laminating. The thickness of TIS is set to half of the electrode separation gap. The density of TIS is 0.15 g cm⁻³.

Comment #7: “Introduction: 1. Li et al. introduced glutaraldehyde to gelatin, forming strong covalent bonds that increased the $T_{h, max}$ from 9°C to 23°C while maintaining $T_{c, min}$ at 21°C. 2. The maximum temperature range of an i-TE liquid cell is limited by the freezing or boiling point of the solvent and the reported $T_{c, min}$ values of i-TE liquid cells always exceed 0°C. These need to be corrected.”

Response: Thanks for the comments. We have corrected the relevant statements in the introduction.

Revision:

(Page 3 in the revised manuscript) “Li *et al.* introduced glutaraldehyde to gelatin, forming strong covalent bonds that increased the $T_{h, \max}$ from 30°C to 44°C while maintaining $T_{c, \min}$ at 21°C.”

(Page 3 in the revised manuscript) “For an i-TE liquid cell, $T_{c, \min}$ is limited by the freezing point of the solution. The $T_{c, \min}$ values of the reported i-TE liquid cells always exceed 0°C.”

Comment #8: “Please check the caption of Figure S5.”

Response: Thanks for your helpful suggestion. The caption of Figure S5 has been corrected.

Revision: (in the revised supporting information) “Figure S5. Thermopower versus concentration of GdmCl for the i-TE liquid cell of H₂O/x FA-y FeCN^{4-/3-}-z GdmCl (x = 0 vol.%, y = 0.4M, z = 0, 1, 2, 3, 4, 5M).”

Comment #9: “4.3 Fabrication of the i-TE liquid cell: A cotton-fiber laminating thermal separator was fabricated by compressing commercial cotton balls with approximate diameters of 20 μm (Figure S7). Please check the figure number.”

Response: Thanks for your helpful suggestion. The figure number has been corrected.

Revision:

(Page 13 in the revised manuscript) “The i-TE liquid cell was assembled with a laminar structure of two graphite current collectors, two CP electrodes, a thermal separator attached to the cold-side electrode, a cylinder spacer composed of

polyformaldehyde (commercial sources), the electrolyte volume at the cell center, and two rubber O-rings (Figure S10).”

(Page 13 in the revised manuscript) “A cotton-fiber laminating thermal separator was fabricated by compressing commercial cotton balls with approximate diameters of 20 μm (Figure S16).”

Reviewer #2

General Comments: *“The authors have revised their manuscript according to the comments. I think it should be accepted for publication.”*

Response: Thank you for your comments. We have revised the introduction part of our paper to include more details about the eutectic solvent strategy and its advantages in the possible application of cryo-temperature region.

REVIEWERS' COMMENTS

Reviewer #1 (Remarks to the Author):

The authors have well revised the manuscript. Therefore, I recommend this manuscript for publication in Nature Communications.